# RSAVQ: Riemannian Sensitivity-Aware Vector Quantization for Large Language Models

**Zukang Xu**[*]
Houmo AI

**Xing Hu**[*]
Houmo AI

**Qiang Wu**
Houmo AI

**Dawei Yang**[†]
Houmo AI

## Abstract

Large language models (LLMs) have demonstrated remarkable performance across a wide range of natural language processing tasks. However, their exponentially increasing parameters pose significant challenges for deployment on resource-constrained devices. Vector Quantization (VQ) shows great promise for low-bit quantization (e.g., 2 to 4 bits), but existing work faces two key challenges: unconstrained direction error and suboptimal bit allocation. In this paper, we propose RSAVQ, a novel VQ framework to enhance extremely low-bit quantization for LLMs. RSAVQ introduces two geometry-driven innovations that effectively mitigate above limitations: (1) Error Direction Sensitivity Guidance (EDSG), which leverages the Fisher Information Matrix (FIM)-induced Riemannian metric to project quantization errors onto low-sensitivity directions in the parameter space. Specifically, this projection is performed along the negative natural gradient direction, which effectively suppresses error expansion. (2) Weight Channel Sensitivity Guidance (WCSG) , which constructs a channel-wise sensitivity metric via FIM curvature analysis to dynamically guide bit resource allocation. The approach facilitates a globally optimal quantization solution within prescribed bit constraints. Experiments demonstrate that RSAVQ outperforms existing methods for LLMs. For example, in 2-bit quantization of LLaMA-3 8B, RSAVQ leads baselines like VPTQ and QuIP# by 0.4 in perplexity (PPL) and 1.5 in zero-shot accuracy. This work offers a practical solution for constrained environments and a theoretical bridge between information geometry and the quantization of neural networks, advancing efficient deep learning.

## 1 Introduction

In recent years, large language models (LLMs) have achieved breakthrough results in natural language processing (NLP), code generation, reasoning, and multimodal tasks[41, 35, 29, 13]. While this progress is impressive, it comes at the cost of a dramatic increase in model size (e.g., the LLaMA-3 70B[35] model demands around 140GB of memory in FP16 precision), posing significant barriers when deploying on resource-constrained devices.

Post-Training Quantization (PTQ)[25] has emerged as a promising technique for reducing the resource footprint of LLMs by converting model weights into lower-bit fixed-point representations without the need for retraining. A typical strategy in PTQ is Scalar Quantization (SQ), where each individual weight is quantized independently to a lower-bit value. Recent work [18, 28, 24, 47, 22] has achieved near-original model accuracy with 4 bit quantization. However, due to the limitations of numerical representation, SQ struggles to maintain performance at extremely low bits (e.g. 3 bit or fewer), often leading to significant degradation in model accuracy.

---

[*]Equal contribution.
[†]Corresponding author: `dawei.yang@houmo.ai`

39th Conference on Neural Information Processing Systems (NeurIPS 2025).

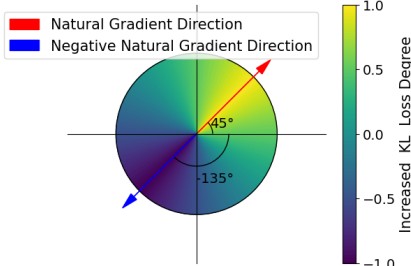
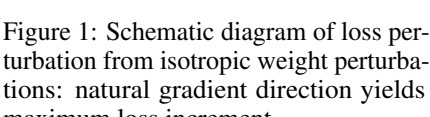

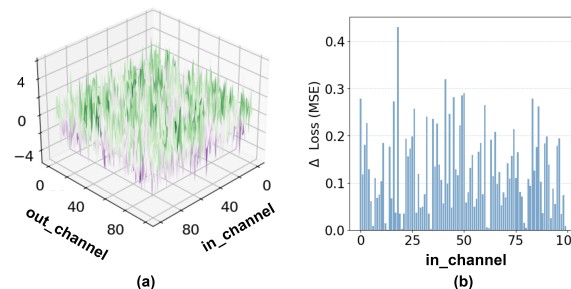

Figure 1: Schematic diagram of loss perturbation from isotropic weight perturbations: natural gradient direction yields maximum loss increment.

Figure 2: (a) Weight distribution of sampling patch in down-project layer of final block in LLaMA-3 8B model. (b) Change in loss after applying the same perturbation( + 0.05 per element) to the input channel of figure (a).

In contrast, Vector Quantization (VQ)[20], another strategy in PTQ shows potential in ultra-low-bit LLM quantization. VQ maps high-dimensional vectors to a set of predefined lower-dimensional vectors and achieves more effective data compression than SQ by leveraging correlations and redundancies across different data dimensions. However, we identify that there are two critical limitations in existing VQ-based methods [42, 44, 31, 14, 50]: (1) **unconstrained direction error**: these methods often overlook directional discrepancies between floating-point vectors and their quantized representations, which significantly affect model performance. In particular, by treating quantization errors as isotropic perturbations under Euclidean assumptions, they fail to capture the geometric relationship between the loss function's sensitivity and the directions of perturbation. As demonstrated in Fig. 1, errors along the negative natural gradient direction ($\theta = -135°$) lead to loss reduction or significantly less loss increase compared to those along the natural gradient direction ($\theta = 45°$). This highlights that error direction control is critical for preserving accuracy. (2) **suboptimal bit allocation**: these methods often assume uniform sensitivity across weight channels and assign equal bit-widths to each channel. However, as shown in Fig. 2(b), applying identical perturbations on each channel results in varying effects on model accuracy, indicating that such a naive bit-allocation policy compromises quantization performance.

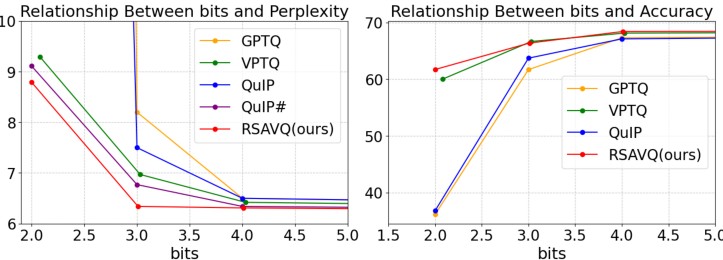

Figure 3: WikiText-2 PPL (left) and average zero-shot accuracy (right) for LLaMA-3 8B quantized at different bit-widths.

To address these limitations, we propose Riemannian Sensitivity-Aware Vector Quantization (RSAVQ), a novel VQ framework that leverages information geometry to model the parameter space (i.e., the weights) of LLMs as a Riemannian manifold with non-uniform curvature—where the local geometry is described by Fisher Information Matrix (FIM) [1, 27]. RSAVQ employs FIM to characterize the local geometric structure of the parameter space, including inter-parameter correlations and manifold curvature, thereby enabling a precise quantification of how parameter perturbations along different directions affect the loss function. RSAVQ comprises two core components: (1)**Error Direction Sensitivity Guidance(EDSG)** projects inevitable quantization errors onto low-sensitivity directions (i.e., along the negative natural gradient directions on the Riemannian manifold), minimizing their adverse impact on model performance and effectively mitigating error accumulation. (2)**Weight Channel Sensitivity Guidance (WCSG)** utilizes FIM to quantify each channel's sensitivity, identifying functionally critical channels and dynamically allocating bit resources to prioritize high-curvature (sensitive) channels while applying aggressive compression to low-sensitivity counterparts. This synergistic design enables RSAVQ to optimize the error direction of each quantized sub-vector through low-sensitivity projection and adaptively allocate bits based on

channel-wise geometric analysis, thus achieving both ultra-low-bit compression and robust accuracy preservation in large language models.

Experiments demonstrate that RSAVQ achieves state-of-the-art performance on LLMs (see Fig. 3 for detailed comparisons). Specifically, for the LLaMA-3 8B model, our method significantly outperforms existing approaches across diverse quantization bit-widths. In particular, under 2-bit quantization, RSAVQ outperforms baselines such as VPTQ and QuIP# by 0.4 in PPL and 1.5 in zero-shot accuracy. Our main contributions are summarized as follows:

- We identify and systematically analyze two critical limitations in vector quantization—unconstrained direction error and suboptimal bit allocation—and demonstrate that they are key contributors of accuracy degradation in ultra-low-bit settings.

- We propose RSAVQ, a novel VQ framework grounded in information geometry. RSAVQ utilizes FIM to guide both error direction projection and per-channel bit allocation, integrating these insights into a single holistic algorithm.

- We validate RSAVQ on LLMs of various model sizes, demonstrating superior performance compared to existing quantization methods, particularly in extremely low-bit scenarios.

## 2 Related Work

Recent PTQ methods can be broadly categorized into scalar and vector quantization. These approaches typically assume a Euclidean parameter space, neglecting the intrinsic geometric structure of deep networks. Although geometric deep learning advances show that the parameter space is better modeled as a non-uniformly curved Riemannian manifold, these insights have only been theoretically explored, with their practical value in extreme low-bit quantization remaining unexplored. Moreover, to date, no unified framework has integrated information geometry to jointly optimize bit allocation and project errors onto low-sensitivity directions. This integration is crucial for minimizing performance degradation. Accordingly, we review related work from two angles: PTQ methods and geometric deep learning research.

### 2.1 PTQ Methods

**Scalar Quantization (SQ),** a classical quantization approach, maps weights to low-bit representations using fixed scaling factors and zero points. It relies on two key assumptions: (1) isotropic errors, where quantization errors in all directions equally impact model performance; (2) uniform parameter sensitivity, treating all parameters as requiring the same bit precision. Methods like GPTQ[18] and AWQ[28] follow this paradigm, with GPTQ incorporating error compensation and AWQ adjusting outlier weights to stabilize quantization. Techniques such as Quarot[3], OSTQuant[23] and MambaQuant[49] utilize Hadamard rotations to enhance weight distribution uniformity, yet these still operate under Euclidean assumptions. Critically, these SQ methods fail to model geometric sensitivities, leading to severe accuracy degradation in extreme low-bit scenarios (e.g. $\leq$ 3-bit) where direction-specific and channel-wise sensitivities dominate error impacts.

**Vector Quantization (VQ)** [20]methods enhance weight compression by clustering weight vectors into shared codebooks. For example, GPTVQ[44] integrates error compensation with EM algorithms to optimize codebooks and indices, while VPTQ[31] and AQLM[14] adopt residual quantization to refine error fitting. CRVQ[48] achieves 1-bit quantization by iteratively selecting critical channels for residual processing, and QuIP# [42] uses Hadamard rotations to preprocess weights before uniform codebook quantization. Despite these advancements, existing VQ techniques lack adaptive bit allocation guided by channel sensitivity and fail to constrain quantization errors to low-impact geometric directions, leading to suboptimal performance in extreme low-bit scenarios.

### 2.2 Geometric Deep Learning

**Geometric Deep Learning** highlights the non-Euclidean nature of neural network parameter spaces, suggesting that they can be more accurately modeled as Riemannian manifolds endowed with FIM[7, 1, 27]. Fisher Information Matrix (FIM)-based approaches[45] define a local Riemannian metric for the parameter space and demonstrate that natural gradient descent achieves faster convergence by adapting to the underlying manifold structure. To address the computational complexity of FIM, methods such as K-FAC[33] propose efficient Kronecker-factored curvature approximations, making second-order optimization feasible for large-scale networks. Beyond optimization, manifold-aware methods[15] have shown that leveraging geometric structures can further unlock hidden information in deep models through Riemannian updates.

**Explorations of Riemannian manifolds in quantization.** Prior works, including CLRQ[6] (manifold geodesic distances for clustering), GLRSQ[39] (strategies for symmetric positive definite matrices), MANIQUEANT[10] (FIM-based gradient mismatch mitigation), FIT[51] (information-geometric metrics for distortion reduction), and PLRQ[40] (probabilistic vector quantization with manifold learning), have explored geometric approaches in quantization. However, these efforts remain fragmented——each focuses on isolated geometric properties (e.g., matrix symmetry, manifold metrics) or individual quantization issues (e.g., gradient alignment, distortion control)——without a unified framework to jointly optimize error direction and channel-wise bit allocation.

## 3 Preliminaries

To facilitate the presentation of our proposed method, we first introduce three key components: Information Geometry and Riemannian Manifolds, Natural Gradient, and Cartesian Product Vector Quantization. Specifically, these concepts form the theoretical foundation of our framework and provide a unified mathematical structure for subsequent sections.

### 3.1 Information Geometry and Riemannian Manifolds

The parameter space of deep neural networks is often assumed to be a high-dimensional Euclidean space. However, this assumption neglects the non-uniform sensitivity of different parameter directions. Recent studies[7, 1, 27, 2] demonstrate that the parameter space can be more accurately modeled as a Riemannian manifold endowed with the Fisher Information Metric[45]. This geometric view better captures the sensitivity of the model to parameter perturbations, providing a theoretical basis for analyzing quantization errors. Let $\mathcal{M}$ denote the differentiable manifold of network parameters $W$. **The Fisher Information Matrix(FIM)**, which we denote by $\mathbf{F}_W$, defines the metric tensor in the parameter space as:

$$F_{ij}(W) = \mathbb{E}\left[\frac{\partial \log p(x|W)}{\partial W_i}\frac{\partial \log p(x|W)}{\partial W_j}\right], \tag{1}$$

where $p(x|W)$ is the model's output distribution. This metric endows the parameter space with a Riemannian geometric structure, and the corresponding **inner product** is induced as follows:

$$\langle \Delta W_1, \Delta W_2 \rangle_W = \Delta W_1^\top \mathbf{F}_W \Delta W_2. \tag{2}$$

This inner product characterizes the geometric distance of parameter perturbations on the manifold, directly relating parameter changes to the KL divergence of the model output distribution (for the derivation process, see Appendix A.5). In practical applications, the channel-wise FIM sub-matrix $\mathbf{F}_c$ is used to approximate the global geometric structure for fine-grained sensitivity analysis.

### 3.2 Negative Natural Gradient

In a flat Euclidean space, the standard negative gradient indicates the direction of the steepest descent. However, on a Riemannian manifold with complex curvature, the optimal descent direction is given by the **negative natural gradient**[1]:

$$-\tilde{\nabla}\mathcal{L} = -\mathbf{F}_W^{-1}\nabla\mathcal{L}. \tag{3}$$

It geometrically corrects the Euclidean gradient through the FIM, ensuring that the parameter update descends along the shortest path (geodesic, defined in Appendix A.6) on the manifold. The negative natural gradient direction $-\tilde{\nabla}\mathcal{L}$ is the direction in which the loss function decreases the fastest, and it is also the "low-sensitivity direction" that the quantization error should try to align with, in order to minimize the impact of the error on the model performance. The derivation of the natural gradient can be based on the Taylor expansion and the Lagrange multiplier method. For the specific derivation steps, refer to Appendix A.4.

### 3.3 Product Vector Quantization

Product Quantization (PQ)[26] extends vector quantization (VQ) by splitting high-dimensional vectors into sub-vectors and quantizing them independently, improving compression efficiency and scalability. Similar to VQ, the optimization process of PQ involves finding the closest cluster center $\mathcal{C}_i$(the $i$-th codeword in the codebook) for an input vector $v$:

$$\arg\min_{i\in k} \|v - \mathcal{C}_i\|^2. \tag{4}$$

For a detailed comparison between PQ and VQ, refer to Appendix A.3. RSAVQ adopts this PQ formulation as its foundation and introduces two key enhancements: specifically, (1) error direction alignment via Riemannian projections, and (2) adaptive bit allocation guided by channel-wise curvature. These modifications jointly improve quantization accuracy under ultra-low-bit settings.

# 4 Method

Traditional quantization methods simplify optimization in Eq. 4 as a Euclidean problem, leading to suboptimal control of quantization error. Inspired by the Riemannian perspective in Preliminaries 3.1—which models the parameter space (i.e. weights) as a curved manifold endowed with the Fisher information metric—we propose the Riemannian Sensitivity-Aware Vector Quantization (RSAVQ) framework. RSAVQ unifies information geometry and vector quantization via two tightly coupled modules: (1) **Error Direction Sensitivity Guidance(EDSG)**, which projects clustering errors onto low-sensitivity (negative natural gradient) directions on the manifold; and (2) **Weight Channel Sensitivity Guidance(WCSG)**, which measures each channel's local curvature to dynamically allocate bits and guide codebook updates. By integrating EDSG and WCSG, RSAVQ minimizes quantization distortion under extreme low-bit constraints while maintaining computational efficiency. For full algorithmic details, see Appendix A.13.

## 4.1 EDSG: Error Direction Sensitivity Guidance

During model quantization, quantization error inevitably occurs, defined as

$$E = W - \mathcal{C}(W) \tag{5}$$

where $\mathcal{C}(\cdot)$ denotes the pseudo-quantization operator. Traditional methods [31] typically rely on mean squared error (MSE) or other Euclidean-based metrics to quantify quantization error and determine optimal coefficients. However, these approaches overlook the geometric sensitivity associated with the direction of the error. Consequently, errors can amplify along directions that are highly sensitive to model performance, resulting in significant accuracy degradation. Although existing techniques [44] attempt to mitigate such errors through post hoc compensation, their initial disregard for directional sensitivity fundamentally limits their effectiveness. To address this limitation, we propose an Error Projection algorithm based on information geometry. Specifically, we project the quantization error onto the negative natural gradient directions of the loss function, which correspond to low-sensitivity directions on the Riemannian manifold. This projection enables a more precise assessment of the quantization error's impact on overall model performance.

As illustrated in Fig. 4, our goal is to project the quantization error onto directions that have minimal influence on the loss function, thereby reducing performance degradation from the perspective of information geometry.

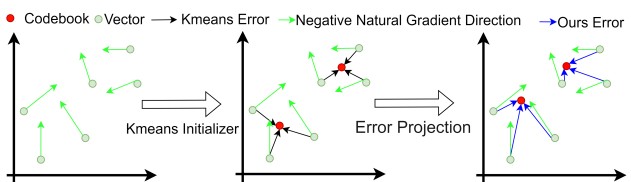

Vector quantization (VQ) has been demonstrated to be a highly promising technique for large model

Figure 4: Clustering process of error projection along negative natural gradient direction.

compression[31, 43], largely due to its inherent flexibility in adjusting the quantization error directions. Building on this property, our objective is to design a codebook $C$ that encourages the quantization error $E$ to lie along "low-sensitivity" directions on the parameter manifold $\mathcal{M}$, thereby minimizing the growth of the KL divergence induced by quantization. Through Lagrangian optimization (see Appendix A.4), we formally prove that projecting $E$ onto the negative natural gradient direction $-\tilde{\nabla}\mathcal{L} = -\mathbf{F}_W^{-1}\nabla\mathcal{L}$ minimizes the first-order increase in the loss function on the Riemannian manifold. Here, $F_W$ denotes the Fisher information matrix, characterizing the local geometry of the parameter space. To enforce this projection during quantization, we define the negative natural gradient direction projection loss in the tangent space $T_W\mathcal{M}$ as:

$$\mathcal{L}_{\text{project}} = \|E + \lambda * \tilde{\nabla}\mathcal{L}\|_{\mathbf{F}}^2 \tag{6}$$

where $\|\cdot\|_F$ denotes the norm induced by the Fisher metric, and $\lambda$ is a hyperparameter that controls the trade-off between strict error projection and quantization flexibility. By minimizing $\mathcal{L}_{\text{project}}$, RSAVQ encourages the quantization error $E$ to remain in low-sensitivity directions, thus preserving model performance even under extremely low-bit quantization.

Building upon the product quantization (PQ) strategy introduced in Preliminaries 3.3, we decompose the weight vectors into a set of sub-vectors $\{v_i\}$, each independently quantized. During the PQ optimization process, we incorporate the projection loss $\mathcal{L}_{\text{project}}$ in the negative natural gradient directions as an additional constraint, alternately optimizing both the codebooks and the assignment indices. This procedure ensures that the quantization error is geometrically projected along the negative natural gradient direction, fundamentally minimizes loss increase and performance degradation However, under a constrained bit budget, different channels exhibit varying tolerances to quantization error. This observation motivates us to further exploit the local curvature information of the parameter manifold to dynamically allocate bit resources, thereby achieving a more fine-grained trade-off between accuracy and compression.

## 4.2 WCSG: Weight Channel Sensitivity Guidance

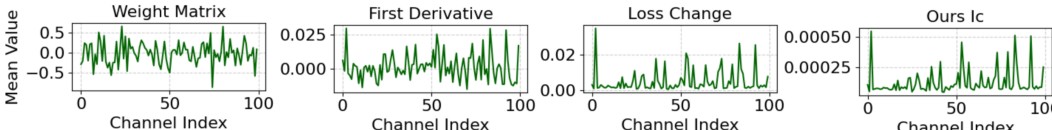

Figure 5: Weight channel sensitivity analysis and comparison.

Building on the channel sensitivity characterization via the Fisher Information Matrix, we propose a geometry-aware bit allocation strategy. Our goal is to further minimize the global quantization distortion in addition to the error direction projection achieved in Method 4.1.

In practice, different channels exhibit heterogeneous sensitivity to quantization perturbations. Thus, a naive uniform bit allocation is suboptimal, and an adaptive strategy is needed to allocate bits proportionally to each channel's importance, thereby minimizing the overall loss degradation.

Existing Euclidean-based sensitivity metrics—such as the gradient magnitude used in SparseGPT[17] or the weight-activation product employed by Wanda[38]—fail to capture the underlying curvature of the parameter space. As demonstrated in Fig. 5, simple statistics like (a) the average weight magnitude or (b) the average gradient per channel show no clear correspondence with (c) the true loss sensitivity under perturbations. This mismatch highlights the necessity of a curvature-aware measurement for accurate sensitivity estimation.

As introduced in Preliminaries 3.1, the negative natural gradient is defined as $-\tilde{\nabla}\mathcal{L} = -\mathbf{F}_W^{-1}\nabla\mathcal{L}$. In traditional Euclidean space, the negative gradient $-\nabla\mathcal{L}$ represents the direction of steepest descent. However, on a Riemannian manifold with non-uniform curvature, the optimal descent direction must be corrected by local metric $\mathbf{F}_W$, leading to the negative natural gradient formulation.

While the negative natural gradient addresses the geometrically optimal update direction, quantifying *weight channel sensitivity* requires a scalar metric that integrates both gradient magnitude and local manifold curvature. To this end, we leverage the Riemannian norm of the negative natural gradient, which measures how parameter perturbations in each channel's tangent space affect the loss function. We decompose the weight tensor $W$ channel-wise as $\{W_c\}$, and for each channel $c$, we compute its Riemannian curvature energy $I_c$ (full derivation in Appendix A.7) as:

$$I_c = \frac{1}{2}|-\tilde{\nabla}\mathcal{L}_c|_W^2 = \frac{1}{2}(\tilde{\nabla}\mathcal{L}_c)^\top \mathbf{F}_c \tilde{\nabla}\mathcal{L}_c. \tag{7}$$

Here, $\mathbf{F}_c$ is the FIM corresponding to channel $c$. The acquisition of FIM is approximated through Kronecker decomposition. For detailed acquisition steps and principles, refer to A.9. Intuitively, $I_c$ captures how sharply the loss landscape curves along the parameters of channel $c$: a larger $I_c$ indicates higher sensitivity to perturbations and thus a greater need for precise quantization.

Under a fixed total bit budget $B_{\max} = \sum b_c$, how to allocate bits across channels becomes crucial for minimizing global quantization distortion. Based on the rate-distortion theory [21], the quantization distortion scales exponentially with the number of bits as $D \propto 2^{-2b}$. Considering the channel curvature sensitivity $I_c$ as an amplification factor for distortion, the global distortion objective can be formulated as:

$$GlobalDistortion = \sum I_c \cdot 2^{-2b_c}. \tag{8}$$

Using Lagrangian optimization (details in Appendix A.8), the optimal bit allocation rule is derived as: $b_c \propto \log_2 I_c$, ensuring that more sensitive channels receive a greater bit allocation.

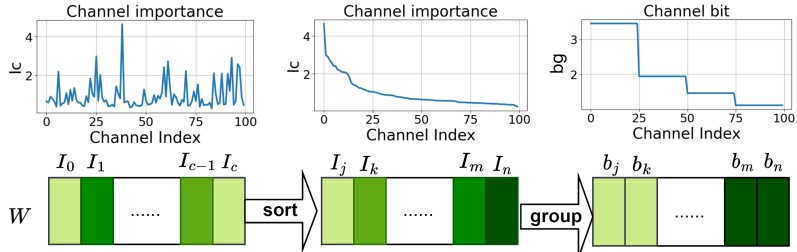

Figure 6: Channel sensitivity-driven channel grouping and bit assignment.

In practice, for a layer with $C$ channels, the bit assignment per channel is given by:

$$b_c = Round\left(B_{\max} \cdot \frac{\log_2 I_c}{\sum_{c=1}^{C} \log_2 I_c}\right). \tag{9}$$

To enable codebook sharing (each group uses a single codebook) while preserving weights channel sensitivity, we introduce a sensitivity-ordered channel grouping strategy. As shown in Fig. 6. First, channels are sorted in descending order of their Riemannian curvature energy $I_c$. We then partition them into $G$ uniform groups, where the group size is $n = \lceil C/G \rceil$ and the $g$-th group contains channels indexed by $G_g = [(g-1)n+1, \min(gn, C)]$. For each group, a unified bit-width $b_g$ is assigned by averaging the dynamically allocated bits of its members, that is,

$$b_g = Round\left(\frac{1}{|G_g|} \sum_{c \in G_g} b_c\right), \tag{10}$$

where $b_c \propto \log_2 I_c$ under the total bit budget constraint. This strategy clusters sensitive channels (high $I_c$) to receive more bits while aggressively compressing less sensitive groups, balancing distortion minimization with practical deployment constraints.

## 5 Experiments

**Baseline.** We focus on weight-only quantization and compare RSAVQ with several strong PTQ baselines, including GPTQ[18], GPTVQ[44], AQLM[14], DB-LLM[9],QuIP[8], QuIP#[42], and VPTQ[31]. Baseline results for these methods are cited from their original papers. For RSAVQ, we use a k-means-based VQ approach similar to VPTQ, with the following settings: the vector length is set to 6, and weight matrices are divided into 4 groups, with each sharing its own codebook. Unless otherwise specified, all experiments are conducted on NVIDIA A100-80GB GPU.

Table 1: The perplexity (ppl) of various quantization algorithms on the LLaMA-2 models when the dataset is Wikitext-2 and the sequence length is 4096, as well as the test performance in various zero-shot tasks.

| Methods | LLaMA-2 7B | | | LLaMA-2 13B | | | LLaMA-2 70B | | |
|---|---|---|---|---|---|---|---|---|---|
| | Bits | W2↓ | 0-shot Avg↑ | Bits | W2↓ | 0-shot Avg↑ | Bits | W2↓ | 0-shot Avg↑ |
| FP16 | 16 | 5.12 | 64.7 | 16 | 4.57 | 67.82 | 16 | 3.12 | 70.21 |
| GPTQ | 2 | 50.75 | 39.16 | 2 | 43.84 | 43.72 | 2 | – | 59.18 |
| GPTVQ | 2.25 | 6.71 | 56.14 | 2.25 | 5.72 | 61.56 | 2.25 | 4.25 | 68.55 |
| DB-LLM | 2.01 | 7.23 | 55.12 | 2.01 | 6.19 | 59.41 | 2.01 | 4.64 | 65.83 |
| AQLM | 2.29 | 6.29 | 58.57 | 2.18 | 5.41 | 61.58 | 2.07 | 3.94 | 68.75 |
| VPTQ | 2.02 | 6.13 | 58.13 | 2.02 | 5.32 | 62.37 | 2.07 | 3.93 | 68.61 |
| QuIP# | 2 | 6.19 | 58.22 | 2 | 5.35 | 61.96 | 2 | 3.91 | 68.94 |
| RSAVQ | 2 | **5.97** | **58.66** | 2 | **5.29** | **62.84** | 2 | **3.55** | **69.05** |
| GPTQ | 3 | 8.06 | 53.1 | 3 | 5.85 | 59.61 | 3 | 4.4 | 65.41 |
| GPTVQ | 3.125 | 5.44 | 62.69 | 3.125 | 4.8 | 59.63 | 3.125 | – | – |
| AQLM | 3.04 | 5.46 | 60.88 | 3.03 | 4.82 | 63.49 | 3.01 | 3.36 | 69.86 |
| VPTQ | 3.02 | 5.43 | 61.72 | 3.03 | 4.79 | 64.21 | 3.01 | 3.34 | 69.58 |
| QuIP# | 3 | 5.41 | – | 3 | 4.78 | – | 3 | 3.35 | – |
| RSAVQ | 3.01 | **5.26** | **62.7** | 3.01 | **4.74** | **66.12** | 3.01 | **3.25** | **70.42** |
| GPTQ | 4 | 5.49 | 60.64 | 4 | 4.78 | 63.87 | 4 | 3.35 | 69.25 |
| GPTVQ | 4.125 | 5.27 | 62.28 | 4.125 | 5.27 | 64.28 | 4.125 | – | – |
| AQLM | 4.04 | 5.21 | 62.54 | 3.94 | 4.65 | 65.12 | 4.14 | 3.19 | 69.93 |
| VPTQ | 4.01 | 5.26 | 61.98 | 4.02 | 4.64 | 64.89 | 4.01 | 3.19 | 69.8 |
| QuIP# | 4 | **5.19** | – | 4 | **4.63** | – | 4 | 3.18 | – |
| RSAVQ | 4.01 | 5.22 | **63.62** | 4.01 | 4.72 | **66.82** | 4.01 | **3.11** | **70.34** |

**Models and Datasets.** We conduct experiments on LLaMA-2 7B, LLaMA-2 13B, LLaMA-2 70B[41], and LLaMA-3 8B, LLaMA-3 70B[35] models to evaluate the performance of our proposed method. The calibration dataset used in our experiments is sampled from the Red_Pajama dataset[46].

To evaluate the performance of the baselines, we compute the perplexity (PPL) of the models on the WikiText-2 dataset[34]. We evaluate the models by randomly sampling sequences from the dataset with the same length as the calibration data. Lower perplexity indicates better preservation of the original output distribution. For direct comparison with methods like VPTQ[31] and Quip#[42], we use the same sequence lengths during testing. Specifically, we test PPL with sequence lengths 4096 for the LLaMA-2 models and 2048 for the LLaMA-3 models.

Additionally, we evaluate generalization capability on several zero-shot tasks, including WinoGrand[37], HellaSwag[52], PIQA[4], ARC-e[5], and ARC-c[12]. All evaluations are performed using the open-source LM-Evaluation-Harness[19] toolkit.

**Main Experimental Results.** Tab. 1 presents the experimental results on the LLaMA-2 series models. We evaluate LLaMA-2 7B, LLaMA-2 13B, and LLaMA-2 70B under a 2-bit quantization configuration. For the LLaMA-2 7B model, RSAVQ achieves a perplexity of 5.97 at 2-bits, which is lower than VPTQ (6.13) and other methods, demonstrating more robust performance on WikiText-2. Furthermore, RSAVQ's zero-shot average accuracy is comparable to other methods and even exceeds them. For the LLaMA-2 70B model, in the 2-bit configuration, RSAVQ's performance is nearly identical to the FP16 baseline, with only a 0.4 PPL

Table 2: PPL on Wikitext-2(seq_len=4096) and 0-shot task accuracy of PTQ algorithms on LLaMA-3.

| Methods | LLaMA-3 8B | | | LLaMA-3 70B | | |
|---|---|---|---|---|---|---|
| | Bits | W2↓ | 0-shot Avg↑ | Bits | W2↓ | 0-shot Avg↑ |
| FP16 | 16 | 6.14 | 68.66 | 16 | 2.9 | 75.32 |
| GPTQ | 2 | 210 | 36.16 | 2 | 11.9 | 45.42 |
| QuIP | 2 | 85.1 | 36.81 | 2 | 13 | 48.66 |
| QuIP# | 2 | 9.11 | – | 2 | 5.6 | – |
| VPTQ | 2.08 | 9.29 | 60.22 | 2.07 | 5.66 | 70.74 |
| RSAVQ | 2 | 8.79 | 61.72 | 2 | 5.6 | 71.3 |
| GPTQ | 3 | 8.2 | 61.7 | 3 | 5.2 | 70.58 |
| QuIP | 3 | 7.5 | 63.72 | 3 | 4.7 | 72.56 |
| QuIP# | 3 | 6.77 | – | 3 | 3.8 | – |
| VPTQ | 3.03 | 6.97 | 66.66 | 3.01 | 3.81 | 73.68 |
| RSAVQ | 3.01 | 6.34 | 66.38 | 3.01 | 3.69 | 74.26 |
| GPTQ | 4 | 6.5 | 67.3 | 4 | 3.3 | 74.88 |
| QuIP | 4 | 6.5 | 67.12 | 4 | 3.4 | 74.52 |
| QuIP# | 4 | 6.34 | – | 4 | 3.21 | – |
| VPTQ | 4.03 | 6.42 | 68.14 | 4.05 | 3.15 | 74.7 |
| RSAVQ | 4.01 | 6.31 | 68.42 | 4.01 | 3.11 | 75.1 |

decrease and a 1.2 accuracy drop, indicating that the model can achieve near original accuracy with significantly reduced bitwidth. Tab. 2 shows the results on LLaMA-3 8B and LLaMA-3 70B models. Under the 2 bit quantization configuration, RSAVQ achieves 61.72 and 71.3 zero-shot performance for the LLaMA-3 8B and LLaMA-3 70B models, respectively. Compared to other methods, such as GPTQ and VPTQ, RSAVQ not only maintains a lower perplexity but also ensures higher zero-shot task accuracy. These results validate the robustness of RSAVQ across different model scales.

**Ablation Study.** To validate the importance of each component in RSAVQ, we performed ablation experiments to compare the performance of the k-means baseline method with the gradual addition of the two components: EDSG and WCSG.The test results are recorded in Tab. 3.

Table 3: On the LLaMA-2 7B model, based on the Wikitext-2 dataset and 0-short datasets, test the effect comparison after using method 1 and method 2 respectively on the original Kmeans basis.

| Bits | Methods | LLaMA-2 7B | | | | | | |
|---|---|---|---|---|---|---|---|---|
| | | W2↓ | AC | AE | HE | QA | WI | Acc Avg↑ |
| | FP16 | 5.12 | 43.3 | 76.3 | 57.1 | 78.1 | 68.7 | 64.70 |
| 2bit | Kmeans | 9.20 | 28.9 | 62.5 | 43.3 | 71.5 | 63.3 | 53.90 |
| | +EDSG | 7.29 (-1.91) | 31.5 | 66.0 | 46.6 | 73.3 | 63.6 | 56.10 (+2.20) |
| | +WCSG | 5.81 (-3.39) | 37.2 | 64.4 | 50.7 | 75.4 | 65.7 | 58.69 (+4.79) |
| 3bit | Kmeans | 7.25 | 35.0 | 68.6 | 47.7 | 73.4 | 64.6 | 57.86 |
| | +EDSG | 5.63 (-1.62) | 40.1 | 72.8 | 53.9 | 76.6 | 66.2 | 61.77 (+3.91) |
| | +WCSG | 5.26 (-1.99) | 41.0 | 73.0 | 54.7 | 76.7 | 68.2 | 62.70 (+4.84) |

On the LLaMA-2 7B model, ablation experiments on various datasets (AC, AE, HE, QA, WI) showed: In the 2-bit quantization setting, the k-means baseline achieved an average accuracy of 53.90. Adding the Error Direction Sensitivity Guidance (EDSG) module improved metrics such as AC,HE, thereby lifting average accuracy to 56.10. Introducing channel grouping further improved performance to 58.69. A similar trend in the 3-bit experiments confirmed that both modules reduce the quantization error and improve accuracy. The results validate their collaborative effect and the framework's effectiveness under extremely low-bit conditions.

We conducted an ablation experiment to evaluate the effect of the hyperparameter $\lambda$ on the performance of RSAVQ. Specifically, we focused on the LLaMA-2 7B model with 2-bit quantization and analyzed the PPL performance on the WikiText-2 dataset for various values of $\lambda$. As shown in Fig. 7. Our experiment showed that as $\lambda$ increased, the quantization accuracy first improved and then decreased. This indicates that $\lambda$ has an optimal range where the projection between quantization error and the natural gradient direction is most effective. Based on our experiments, we found that the optimal range for $\lambda$ lies between 0.01 and 0.1.Additional ablation studies across multiple models under 2-bit quantization are reported in Appendix A.10, showing that $\lambda$ values in the range [0.01, 0.1] are robust across

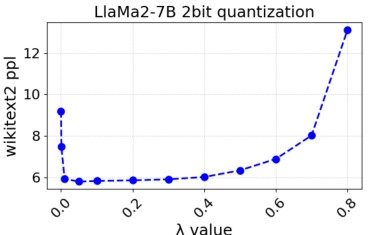

Figure 7: Optimal $\lambda$ Tuning in 2-bit quantized LLaMA-2 7B.

architectures. This result highlights the importance of tuning $\lambda$ for balancing the projection strength between quantization error and natural gradient direction, with the optimal value providing the best trade-off between error reduction and model accuracy. We further analyze the sensitivity of RSAVQ to group size and codebook vector length in Appendix A.11.

**Speed and Efficiency Testing.** In terms of hardware efficiency, we conducted speed and memory usage tests on the LLaMA-2 7B and LLaMA-2 13B models running inference on a single NVIDIA A100 GPU. The results demonstrate the significant improvements in inference speed and memory efficiency achieved by RSAVQ under low-bit quantization.

As shown in Tab. 4, for the LLaMA-2 7B model, the inference speed with the FP16 model was 38.6 tokens/s, while under 2-bit quantization, the speed achieved a 1.57x speedup. Similarly, the LLaMA-2 13B model achieved comparable acceleration under 2-bit quantization. These results highlight

Table 4: Speed and Memory Testing

| Bits | LLaMA-2 7B | | LLaMA-2 13B | |
|------|------------|-----------|-------------|-----------|
| | tokens/s | memory(G) | tokens/s | memory(G) |
| FP16 | 38.60 | 13.16 | 24.29 | 25.42 |
| 2 | 60.60 | 2.28 | 44.21 | 4.03 |
| 3 | 50.57 | 3.83 | 29.15 | 5.56 |

that RSAVQ not only maintains low memory usage but also provides a significant speedup in inference, making it highly suitable for edge devices or resource-constrained environments where both memory and computation resources are limited. Beyond inference efficiency, RSAVQ also reduces the offline quantization time, as detailed in AppendixA.12.

RSAVQ achieves superior low-bit performance by integrating information geometry and channel-wise bit allocation, outperforming traditional methods in PPL, zero-shot accuracy, and hardware efficiency across LLaMA models.

## 6 Conclusion

This paper introduces RSAVQ, a Riemannian geometry-driven VQ framework that addresses extreme low-bit quantization challenges in LLMs. RSAVQ features two key innovations: (1) Error Direction Sensitivity Guidance (EDSG), which projects quantization errors onto low-sensitivity directions (negative natural gradient) using the Fisher Information Matrix (FIM), minimizing performance degradation; (2) Weight Channel Sensitivity Guidance (WCSG), which dynamically allocates bit resources based on FIM-derived curvature metrics to prioritize sensitive channels, balancing accuracy and compression efficiency.

Theoretically, RSAVQ bridges information geometry and neural network quantization by modeling the parameter space as a Riemannian manifold, enabling geometrically informed error control and adaptive bit allocation. The proposed Riemannian curvature energy metric $I_c$ offers a principled way to quantify channel sensitivity, overcoming Euclidean-based limitations. Practically, its hardware-friendly channel grouping strategy ensures efficient inference while achieving state-of-the-art 2-bit compression.

Experimental results on LLaMA models demonstrate RSAVQ's superiority, particularly in extreme low-bit scenarios (e.g., 2-bit quantization with minimal PPL degradation and high zero-shot accuracy). By unifying geometric insights with quantization techniques, this work provides a robust solution for deploying LLMs in resource-constrained environments and opens new avenues for integrating information geometry into model optimization. While RSAVQ exhibits strong empirical performance, its limitations——including limited cross-architecture and cross-domain generality, unexplored scalar quantization applicability, and hardware-specific memory efficiency constraints——are analyzed in detail in Appendix A.1.

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

# A Appendix

## A.1 Limitations

Despite its strong empirical gains, RSAVQ has three main limitations that merit further study:

- **Domain and architecture generality.** RSAVQ is validated primarily on Transformer-based language models (e.g., LLaMA), with limited empirical exploration in multimodal models, computer vision architectures, or reinforcement learning frameworks.
- **Quantization method scope.** This work focuses exclusively on applying information geometry to vector quantization (VQ), while its potential utility in scalar quantization (SQ) remains unexplored. Extending geometric sensitivity analysis to SQ could reveal new optimization strategies for low-bit quantization.
- **Hardware deployment efficiency.** Vector quantization's reliance on codebook lookups introduces memory access overhead, particularly on heterogeneous hardware (CPUs/GPUs/edge devices), with limited optimization for platform-specific memory systems (e.g., cache-friendly indexing, parallel computation).

Future work will explore cross-architecture generalization, scalar quantization extensions, and hardware-aware quantization optimizations to enhance RSAVQ's practical applicability.

## A.2 Acknowledgments

We would like to acknowledge the assistance of large language models in improving the clarity and readability of the manuscript text. We emphasize that all experiments, analyses, and core methodological innovations presented in this paper were independently conceived and executed by the authors, without the use of large language models for generation.

We are also grateful to the reviewers, the area chairs and the program chairs for their valuable time and constructive feedback during the review process, which helped us significantly improve the quality of this work.

## A.3 Vector Quantization vs. Product Quantization

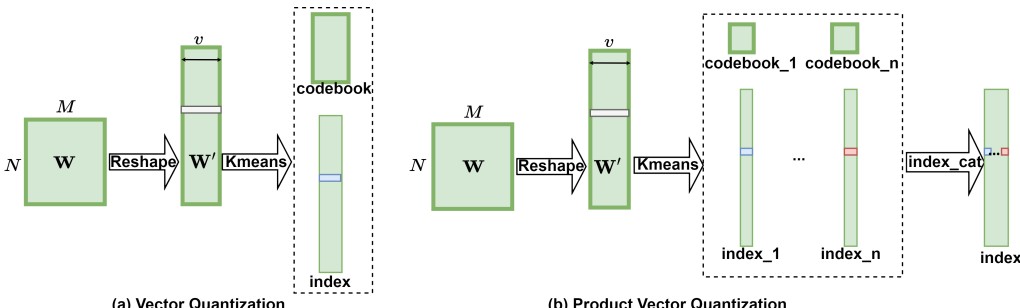

Figure 8: Schematic diagram of standard vector quantization (a) and product vector quantization (b).

**Vector Quantization (VQ).** As shown in Fig. 8(a), given a weight matrix $W \in \mathbb{R}^{M \times N}$, we first reshape it into a sequence of $v$-dimensional vectors $W' \in \mathbb{R}^{\frac{MN}{v} \times v}$. A codebook $\mathcal{C} \in \mathbb{R}^{k \times v}$ is then built using $k$-means clustering.

Each vector in $W'$ is quantized by selecting the nearest centroid using:

$$\arg \min_{i \in \{1,\dots,k\}} \|v - \mathcal{C}_i\|^2, \tag{11}$$

where $\mathcal{C}_i$ denotes the $i$-th codeword. The quantized results are stored as an index array referencing the codebook entries.

**Product Quantization (PQ).** To scale quantization to high-dimensional vectors, PQ divides each $v$-dimensional vector into $n$ equal-length sub-vectors:

$$v = [v_1, v_2, \ldots, v_n], \quad v_i \in \mathbb{R}^{v/n}. \tag{12}$$

Each sub-vector $v_i$ is independently quantized using a dedicated codebook $\mathcal{C}_i \in \mathbb{R}^{k \times v/n}$, producing $n$ sub-indices. These indices are concatenated into a single composite index that jointly encodes the original vector.

## A.4 Principle of Negative Natural Gradient

Referring to[32], the derivation process is as follows:

Let the small change of the loss function $L(W)$ at point $W$ be $\Delta L$. According to the first-order Taylor expansion, we have:

$$\Delta L \approx \nabla L^\top \Delta W \tag{13}$$

On the Riemannian manifold, we hope to find a direction $\Delta W$ so that the loss function decreases the fastest under the given Riemannian metric. According to the Lagrange multiplier method, introduce the constraint condition $\langle \Delta W, \Delta W \rangle_W = 1$, and construct the Lagrange function:

$$L(\Delta W, \lambda) = \nabla L^\top \Delta W - \lambda(\Delta W^\top F_W \Delta W - 1) \tag{14}$$

Take the partial derivative of $\Delta W$ and set it to zero to obtain:

$$\nabla L - 2\lambda F_W \Delta W = 0 \tag{15}$$

Get $\Delta W = \frac{2}{\lambda} F_W^{-1} \nabla L$. Let $-\widetilde{\nabla} L = -F_W^{-1} \nabla L$, which is the negative natural gradient.

## A.5 Detailed Derivation of KL Divergence Approximation

The following presents the detailed derivation of the KL divergence approximation formula, involving Taylor expansion, the definition of the Fisher Information Matrix (FIM), and differential properties of probabilistic models:

For parameterized probability distributions $p(x|W)$ and $p(x|W + \Delta W)$, the KL divergence is defined as:

$$D_{\text{KL}}(p(x|W) \,\|\, p(x|W + \Delta W)) = \mathbb{E}_{x \sim p(x|W)} \left[ \log \frac{p(x|W)}{p(x|W + \Delta W)} \right]. \tag{16}$$

Expanding the logarithmic term:

$$D_{\text{KL}} = \mathbb{E} \left[ \log p(x|W) - \log p(x|W + \Delta W) \right]. \tag{17}$$

Let $f(W) = \log p(x|W)$. For a sufficiently small parameter perturbation $\Delta W$, the second-order Taylor expansion of $f(W + \Delta W)$ around $W$ (neglecting higher-order terms) is:

$$f(W + \Delta W) \approx f(W) + \nabla f(W)^\top \Delta W + \frac{1}{2} \Delta W^\top \mathbf{H}_f(W) \Delta W, \tag{18}$$

where: - $\nabla f(W) = \frac{\partial \log p(x|W)}{\partial W}$ is the gradient (score function), - $\mathbf{H}_f(W) = \frac{\partial^2 \log p(x|W)}{\partial W \partial W^\top}$ is the Hessian matrix.

Substituting into the KL divergence expression:

$$D_{\text{KL}} \approx \mathbb{E} \left[ f(W) - \left( f(W) + \nabla f^\top \Delta W + \frac{1}{2} \Delta W^\top \mathbf{H}_f \Delta W \right) \right] = \mathbb{E} \left[ -\nabla f^\top \Delta W - \frac{1}{2} \Delta W^\top \mathbf{H}_f \Delta W \right]. \tag{19}$$

By the **property of the score function** (regularity condition of probability distributions):

$$\mathbb{E} \left[ \nabla \log p(x|W) \right] = \int p(x|W) \cdot \nabla \log p(x|W) \, dx = \nabla \int p(x|W) \, dx = \nabla 1 = \mathbf{0}. \tag{20}$$

Thus, the expectation of the first-order term vanishes:

$$\mathbb{E} \left[ -\nabla f^\top \Delta W \right] = -\Delta W^\top \mathbb{E} \left[ \nabla f \right] = \mathbf{0}. \tag{21}$$

Using the **Information Matrix Equality**:

$$\mathbb{E}\left[\mathbf{H}_f(W)\right] = -\mathbb{E}\left[\nabla \log p(x|W) \nabla \log p(x|W)^\top\right] = -\mathbf{F}_W, \tag{22}$$

where $\mathbf{F}_W$ is the Fisher Information Matrix (FIM), defined as:

$$\mathbf{F}_W = \mathbb{E}\left[\nabla \log p(x|W) \nabla \log p(x|W)^\top\right]. \tag{23}$$

The expectation of the second-order term becomes:

$$\mathbb{E}\left[-\frac{1}{2}\Delta W^\top \mathbf{H}_f \Delta W\right] = \frac{1}{2}\Delta W^\top \mathbb{E}\left[-\mathbf{H}_f\right]\Delta W = \frac{1}{2}\Delta W^\top \mathbf{F}_W \Delta W. \tag{24}$$

Under the Riemannian metric defined in the paper, the inner product is:

$$\langle \Delta W_1, \Delta W_2 \rangle_W = \Delta W_1^\top \mathbf{F}_W \Delta W_2, \tag{25}$$

so the second-order term can be expressed as:

$$\frac{1}{2}\Delta W^\top \mathbf{F}_W \Delta W = \frac{1}{2}\langle \Delta W, \Delta W \rangle_W. \tag{26}$$

Neglecting higher-order infinitesimal terms, the KL divergence approximates to:

$$D_{\mathrm{KL}}(p(x|W) \,\|\, p(x|W + \Delta W)) \approx \frac{1}{2}\langle \Delta W, \Delta W \rangle_W. \tag{27}$$

This approximation reveals the direct relationship between parameter perturbations in the Riemannian manifold (measured by FIM) and changes in the model's output distribution (KL divergence).

In practical applications, to reduce computational overhead or enable fine-grained analysis, block-wise approximations (such as diagonal blocks or K-FAC approximation[33]) are often used to decompose the global Fisher matrix into channel-level (or subnetwork-level) submatrices $\mathbf{F}_c$. These submatrices provide local geometric information for each channel, laying the foundation for subsequent sensitivity analysis and bit allocation.

### A.6 Geodesic

In Riemannian manifolds, the shortest path between two points is called a geodesic[11], and its length is defined as the geodesic distance. For two points $p, q \in M$ on the manifold, their **geodesic distance** definition is:

$$d(p, q) = \int_0^1 \|\dot{\gamma}(t)\| \, dt. \tag{28}$$

Here, $\gamma(t)$ is the geodesic that connects $p$ and $q$, and $\dot{\gamma}(t)$ is the geodesic velocity vector. Geodesic distance is an important measure of the relationship between points on a manifold. Especially in quantitative error analysis, it can help us evaluate the impact of quantization error on model performance. For example, in our method, the direction of quantization error is projected onto a low-sensitivity direction to minimize its contribution to the geodesic distance. If the manifold is compared to the surface of the earth, then the geodesic distance is the length of the great circular arc between two points. In the parameter space of the neural network, the geodesic distance on the manifold based on the $F_W$ measure in Preliminaries 3.1 reflects the actual impact of the parameter changes on the performance of the model.

### A.7 Riemannian Curvature Energy

For each channel $c$, we aim to measure the impact of parameter perturbations on the loss function. In deep learning, methods relying on the magnitude of weight gradients to determine weight importance have long dominated, as the gradient magnitude directly reflects the dynamic contribution of weights to the loss function. Early studies [16] proposed using the L2 norm of weights to assess importance, but subsequent work revealed that gradient information can more accurately capture the real-time impact of weights. For example, [30, 53] introduced the use of the L1 norm of gradients to filter important weights, and [36] further incorporated gradient direction information by fusing L1 and L2 norms for weight scoring. However, directly using the Euclidean norm of gradients $\|\nabla \mathcal{L}_c\|$ overlooks

a critical fact: the parameter space of deep neural networks is inherently a Riemannian manifold. Therefore, instead of directly using gradients, we employ the Riemannian norm of the negative natural gradient:

$$\| - \tilde{\nabla}\mathcal{L}_c \|_W^2 = \tilde{\nabla}\mathcal{L}_c^\top \mathbf{F}_c \tilde{\nabla}\mathcal{L}_c, \tag{29}$$

where the negative natural gradient $-\tilde{\nabla}\mathcal{L}_c = -\mathbf{F}_c^{-1}\nabla\mathcal{L}_c$ projects the Euclidean gradient onto the tangent space of the Riemannian manifold. Substituting the negative natural gradient into the norm yields:

$$\| - \tilde{\nabla}\mathcal{L}_c \|_W^2 = \left( \mathbf{F}_c^{-1}\nabla\mathcal{L}_c \right)^\top \mathbf{F}_c \left( \mathbf{F}_c^{-1}\nabla\mathcal{L}_c \right) = (\nabla\mathcal{L}_c)^\top \mathbf{F}_c^{-1}\nabla\mathcal{L}_c. \tag{30}$$

We thus define the **Riemannian curvature energy** as:

$$I_c = \frac{1}{2}(\nabla\mathcal{L}_c)^\top \mathbf{F}_c^{-1}\nabla\mathcal{L}_c. \tag{31}$$

Physical and Geometric Interpretation The formula can be interpreted through two key perspectives: (1) Balance between Gradient and Metric Scaling: The numerator $\nabla\mathcal{L}_c$ represents the gradient magnitude of channel $c$, indicating the local sensitivity of the loss function to parameter changes. The matrix $\mathbf{F}_c^{-1}$, the inverse of the Fisher information matrix, characterizes the *local metric scaling* of the parameter space. When the eigenvalues of $\mathbf{F}_c$ are large in a particular direction (signifying high local curvature or strong parameter correlations), the corresponding components of $\mathbf{F}_c^{-1}$ are small, which suppresses the gradient contribution in that direction. This mechanism avoids misjudgments of importance caused by geometric distortion in the parameter space. In contrast, the Euclidean norm implicitly assumes $\mathbf{F}_c = \mathbf{I}$ (the identity matrix), ignoring the actual geometric structure of the manifold.

(2) Impact of Unit Geometric Perturbations: The Riemannian norm of the negative natural gradient measures the effect of *unit-length perturbations in the Riemannian manifold* on the loss function. By normalizing with $\mathbf{F}_c^{-1}$, perturbations in different parameter directions are converted to the manifold's intrinsic geometric scale. In high-curvature directions (where $\mathbf{F}_c$ has large eigenvalues), the same Euclidean-distance perturbation corresponds to a smaller effective geometric distance in the manifold, leading to a weaker impact on the loss. This property enables $I_c$ to accurately capture the dynamic importance of channel parameters in the true geometric space, overcoming the scale biases inherent in the Euclidean framework.

### A.8   Bit Allocation Principle

Following the inference from [21], the quantization error can be approximated as uniformly distributed, and the quantization distortion $D$ has an exponential relationship with the number of quantization bits $b$, expressed as:

$$D \propto 2^{-2b}. \tag{32}$$

In our framework, for each channel $W_c$, the quantization distortion is influenced not only by the number of quantization bits but also by the sensitivity of the channel to the loss function. This sensitivity is measured by $I_c \geq 1$ (where we ensure non-negativity by shifting $I_c \leftarrow I_c + 1$). Intuitively, $I_c$ reflects that even with the same magnitude of quantization error, the impact on the final loss can vary: a larger $I_c$ indicates that the channel has stronger gradients and higher curvature, meaning the quantization error is amplified and causes greater loss degradation; conversely, a smaller $I_c$ suggests the channel is less sensitive to errors, resulting in smaller loss impact.

Therefore, the quantization distortion for channel $c$ can be formulated as:

$$D_c(b_c) \propto I_c \cdot 2^{-2b_c}. \tag{33}$$

Under the global bit budget constraint $\sum_{c=1}^C b_c = B_{\max}$, where $B_{\max}$ denotes the total bit allocation across all channels (e.g., $B_{\max} = 2 \times C$ for 2-bit quantization), we aim to minimize the overall distortion. To this end, we construct the following Lagrangian:

$$\mathcal{L}(\{b_c\}, \gamma) = \sum_{c=1}^C I_c \cdot 2^{-2b_c} + \gamma \left( \sum_{c=1}^C b_c - B_{\max} \right). \tag{34}$$

Taking the partial derivative of $\mathcal{L}$ with respect to each $b_c$ and setting it to zero (ignoring discretization effects), we obtain:

$$\frac{\partial \mathcal{L}}{\partial b_c} = -2\ln 2 \cdot I_c \cdot 2^{-2b_c} + \gamma = 0. \tag{35}$$

which leads to:

$$2^{-2b_c} = \frac{\gamma}{2 \ln 2 \cdot I_c}. \tag{36}$$

Taking the logarithm on both sides, we further derive:

$$b_c \propto \log_2(I_c). \tag{37}$$

Considering the overall bit budget $B_{\max}$, the final bit allocation formula is:

$$b_c = B_{\max} \cdot \frac{\log_2 I_c}{\sum_{c=1}^{C} \log_2 I_c}. \tag{38}$$

## A.9 Approximation of FIM

Given the extremely high complexity of directly computing the full FIM (scaling with the square of the weight dimension), we adopt Kronecker factorization approximation (instead of diagonalization) to decompose the FIM into the tensor product of two low-dimensional matrices: $F \approx F_O \otimes F_I$, where $F_O \in \mathbb{R}^{m \times m}$ (output channel FIM) and $F_I \in \mathbb{R}^{n \times n}$ (input channel FIM). This decomposition is implemented as follows:

1. Gradient definition: $\nabla_W \ell$ denotes the gradient of the loss function with respect to the weight $W$, computed for a single sequence $s$.

2. Calculation of $F_I$: Input channel statistics are estimated via the expectation of the outer product of gradients:
$$F_I = \frac{1}{m} \cdot \mathbb{E}_{s \sim \mathcal{D}} \left[ (\nabla_W \ell)^T \cdot (\nabla_W \ell) \right].$$

3. Calculation of $F_O$: Output channel statistics are estimated via the expectation of the outer product of gradients:
$$F_O = \frac{1}{n} \cdot \mathbb{E}_{s \sim \mathcal{D}} \left[ (\nabla_W \ell) \cdot (\nabla_W \ell)^T \right].$$

## A.10 Additional Experiments on the Projection Hyperparameter $\lambda$

We conducted further ablation studies to examine the sensitivity of the projection hyperparameter $\lambda$ across different model architectures under the 2-bit quantization setting on the Wikitext2 test set. Specifically, we tested three representative models: LLaMA2-7B, LLaMA3-8B, and Qwen2.5-7B. The results are summarized in Table 5.

Overall, we find that $\lambda$ values in the range $[0.01, 0.1]$ consistently yield strong performance across all tested models. Based on these results, we recommend setting $\lambda = 0.05$ as a robust default choice.

Table 5: Ablation study of projection hyperparameter $\lambda$ under 2-bit quantization on Wikitext2. Numbers denote perplexity (lower is better).

| Model | 0 | 0.001 | 0.01 | 0.05 | 0.1 | 0.2 | 0.4 | 0.6 | 0.8 |
|-------|-----|-------|------|------|------|-------|-------|-------|-------|
| LLaMA2-7B | 9.20 | 7.51 | 5.94 | 5.81 | 5.84 | 5.87 | 6.03 | 6.90 | 13.10 |
| LLaMA3-8B | 13.32 | 11.19 | 9.17 | 8.79 | 8.78 | 8.99 | 14.86 | 17.63 | 20.92 |
| Qwen2.5-7B | 13.17 | 10.37 | 8.75 | 8.77 | 9.02 | 11.27 | 15.90 | 19.71 | 36.74 |

## A.11 Ablation Studies on Group Size and Codebook Vector Length

We provide additional ablation studies to analyze the sensitivity of RSAVQ to two critical implementation parameters: the number of groups in WCSG and the codebook vector length. These results complement the main text and justify our chosen default configurations.

**Effect of group size**  We conducted experiments on the LLaMA2-7B model with WikiText2 (sequence length 4096, 2-bit quantization, vector length 6). Results in Table 6 show that performance improves as the number of groups increases, but the gain diminishes after 4 groups. Specifically, perplexity (ppl) drops significantly from 6.03 to 5.81 when increasing groups from 2 to 4, while further increases (6–10 groups) yield stable results around 5.78–5.79. This indicates moderate sensitivity to group size, with stable performance at $\geq 4$ groups. We thus adopt 4 groups as the default configuration to balance performance and efficiency.

Table 6: Perplexity (ppl) of LLaMA2-7B with different WCSG group sizes under 2-bit quantization.

| Groups | 2 | 3 | 4 | 6 | 8 | 10 |
|--------|------|------|------|------|------|------|
| ppl | 6.03 | 5.88 | 5.81 | 5.79 | 5.78 | 5.78 |

**Effect of codebook vector length**  Vector length plays a central role in vector quantization. We analyzed its impact using LLaMA2-7B on WikiText2 (sequence length 4096, 2-bit quantization, 2 groups for product quantization, 4 groups for WCSG). Table 7 shows that longer vector lengths yield slight performance gains (e.g., ppl decreases from 5.81 at length 6 to 5.62 at length 14). However, increasing vector length also raises storage and quantization costs. For example, at 2-bit quantization, vector length growth from 6 to 14 increases the average bit count from 2.0 to 2.875, with corresponding bandwidth costs.

This demonstrates that RSAVQ's sensitivity to vector length lies in the performance–cost trade-off: longer vectors offer better performance but incur higher costs. To balance accuracy and efficiency, we use a vector length of 6 in main experiments.

Table 7: Impact of codebook vector length on perplexity (ppl) and average bit count under 2-bit quantization (LLaMA2-7B, WikiText2).

| Vector length | 4 | 6 | 8 | 10 | 12 | 14 |
|---------------|------|------|------|------|------|------|
| Avg. bits | 2.00 | 2.00 | 2.00 | 2.04 | 2.19 | 2.88 |
| ppl | 5.97 | 5.81 | 5.81 | 5.81 | 5.75 | 5.62 |

The above results confirm that RSAVQ is moderately sensitive to group size and vector length. Performance saturates after 4 groups, and vector length offers a tunable trade-off between accuracy and cost. The chosen defaults (group size = 4, vector length = 6) provide a balanced configuration for our main experiments.

## A.12  Quantization Time Comparison

We have supplemented data on the efficiency of the offline quantization process, comparing RSAVQ with mainstream PTQ methods (VPTQ, GPTVQ) under a 2-bit configuration on an 80GB A100 GPU. Our method adopts a product quantization scheme, which not only reduces quantization time and codebook size compared to conventional vector quantization, but also achieves quantization latency comparable to GPTVQ, where the $vector\_length = 1$.

Table 8: Quantization time comparison of different methods on LLaMA2.

| | LLaMA2-7B | LLaMA2-13B | LLaMA2-70B |
|-------|-------------|-------------|-------------|
| VPTQ | 2 GPU hours | 3.5 GPU hours | 19 GPU hours |
| GPTVQ | 1 GPU hours | 1.8 GPU hours | 8 GPU hours |
| RSAVQ | 1.2 GPU hours | 2 GPU hours | 10 GPU hours |

## A.13 Algorithm

---

**Algorithm 1** RSAVQ: Riemannian Sensitivity-Aware Vector Quantization

---

**Input:** Original weight matrix $W \in \mathbb{R}^{M \times N}$, total bit budget $B_{\max}$, sub-vector length $v$, iterations $T$, small positive constants $\lambda$, number of groups $G$, Fisher matrices $\{\mathbf{F}_c\}_{c=1}^{C}$

**Output:** Quantized codebook $\mathcal{C}$, index matrix $\mathcal{I}$, original channel positions $\mathcal{P}$

---

**CWSG: Channel-wise Sensitivity Guidance**

1: Initialize: $\mathcal{P}[c] = c$ (preserve original channel order)
2: Compute channel importance scores using Riemannian metric:
3: **for** $c = 1$ to $C$ **do**
4:     Compute natural gradient: $\tilde{\nabla}\mathcal{L}_c = \mathbf{F}_c^{-1}\nabla\mathcal{L}_c$                    ▷ Eq 3 in paper
5:     Importance score: $I_c = \frac{1}{2}\tilde{\nabla}\mathcal{L}_c^{\top}\mathbf{F}_c\tilde{\nabla}\mathcal{L}_c$          ▷ Fisher-weighted sensitivity, Eq 7 in paper
6:     Store $I_c$ in $\mathcal{I}_{importance}[c]$
7: **end for**
8: Sort channels by $\mathcal{I}_{importance}$ (descending) and reorder $W, \mathcal{P}$
9: Allocate bits proportionally to sensitivity:
10: $I[c] = I[c] + 1$
11: **for** $c = 1$ to $C$ **do**
12:     $b[c] = \text{Round}\left(B_{\max} \cdot \frac{\log_2 I[c]}{\sum_{c=1}^{C}\log_2 I[c]}\right)$                    ▷ Eq 9 in paper
13: **end for**
14: Channel grouping: Divide sorted channels into $G$ groups with equal size
15: $n = \lceil C/G \rceil$
16: **for** $g = 1$ to $G$ **do**
17:     Group $g$ channels: $G_g = [(g-1)n + 1, \min(gn, C)]$
18:     Average bits per group: $b_g = \text{Round}\left(\frac{\sum_{c \in G_g} b[c]}{|G_g|}\right)$                    ▷ Eq 10 in paper
19: **end for**

---

**EDSG: Error Direction Sensitivity Guidance**

20: **for** $g = 1$ to $G$ **do**
21:     Extract group weights and reshape into sub-vectors: $W_g \in \mathbb{R}^{M \times |G_g|} \to \{v_{g,l} \in \mathbb{R}^v\}$
22:     Initialize codebook $\mathcal{C}_g$ via K-means with $2^{b_g[g]}$ cluster centers
23:     **for** $t = 1$ to $T$ **do**
24:         **for** each sub-vector $v_{g,l}$ **do**
25:             Quantize: $i_{g,l} = \arg\min_i \|v_{g,l} - \mathcal{C}_{g,i}\|^2$
26:             Compute error: $E_{g,l} = v_{g,l} - \mathcal{C}_{g,i_{g,l}}$
27:             Project error to low-sensitivity direction:
28:             $\mathcal{L}_{\text{project}} \leftarrow \|E_{g,l} + \lambda \cdot \tilde{\nabla}\mathcal{L}\|_{\mathbf{F}}^2$                    ▷ Eq 6 in paper
29:             Update $\mathcal{C}_g$ via gradient descent on $\mathcal{L}_{\text{project}}$
30:         **end for**
31:         Store index $\mathcal{I}[g, l] = i_{g,l}$
32:     **end for**
33: **end for**
34: Reconstruct $\hat{W}$ from $\mathcal{C}$ and $\mathcal{I}$
35: **return** $\mathcal{C}, \mathcal{I}, \mathcal{P}$

---

## A.14  Additional Experiments

In this section we report additional experimental results for LLaMA-2 models[41] and LLaMA-3 models[35].

Table 9: Perplexity on Wikitext-2(Sequence Length=4096) and Zero-Shot Task Accuracy of Various Quantization Algorithms on LLaMA-2 7B.

| LlaMA-2 7B seqlen=4096 | bit | W2↓ | AC | AE | HE | QA | WI | Avg↑ |
|---|---|---|---|---|---|---|---|---|
| FP16 | 16 | 5.12 | 43.3 | 76.3 | 57.1 | 78.1 | 68.7 | 64.7 |
| GPTQ | 2 | 50.75 | 20.9 | 34.9 | 30.5 | 57.2 | 52.3 | 39.16 |
| GPTVQ | 2.25 | 6.71 | 31.2 | 66.3 | 46.4 | 72.4 | 64.4 | 56.14 |
| DB-LLM | 2.01 | 7.23 | 33.53 | 45.2 | 61.98 | 73.18 | 61.72 | 55.12 |
| AQLM | 2.29 | 6.29 | 34.9 | 66.5 | 50.88 | 74.92 | 65.67 | 58.57 |
| VPTQ | 2.02 | 6.13 | 35.24 | 63.8 | 52.08 | 75.19 | 64.33 | 58.13 |
| QuIP# | 2 | 6.19 | 34.6 | 64.6 | 51.91 | 75.1 | 64.9 | 58.22 |
| RSAVQ(ours) | 2 | 5.81 | 37.2 | 64.4 | 50.71 | 75.4 | 65.74 | 58.6 |
| GPTQ | 3 | 8.06 | 31.1 | 58.5 | 45.2 | 71.5 | 59.2 | 53.1 |
| GPTVQ | 3.125 | 5.44 | 39.93 | 74.07 | 54.21 | 76.17 | 69.06 | 62.69 |
| AQLM | 3.04 | 5.46 | 38.4 | 68.06 | 54.12 | 76.88 | 66.93 | 60.88 |
| VPTQ | 3.02 | 5.43 | 39.3 | 69.1 | 54.9 | 77.3 | 68 | 61.72 |
| QuIP# | 3 | 5.41 | 39.2 | 68.4 | – | 77.3 | 66.5 | – |
| RSAVQ(ours) | 3.01 | 5.26 | 41 | 73 | 54.7 | 76.7 | 68.2 | 62.7 |
| GPTQ | 4 | 5.49 | 36.8 | 66.2 | 55.4 | 76.6 | 68.2 | 60.64 |
| GPTVQ | 4.125 | 5.27 | 42.83 | 75.17 | 56.41 | 77.37 | 69.61 | 62.28 |
| AQLM | 4.04 | 5.21 | 41 | 70.2 | 56 | 78.2 | 67.3 | 62.54 |
| VPTQ | 4.01 | 5.26 | 39.7 | 69 | 56 | 78.1 | 67.1 | 61.98 |
| QuIP# | 4 | 5.19 | 40.5 | 69.1 | – | 78.4 | 67.6 | – |
| RSAVQ(ours) | 4.01 | 5.22 | 42 | 74.7 | 56.3 | 76.9 | 68.2 | 63.62 |

Table 10: Perplexity on Wikitext-2(Sequence Length=4096) and Zero-Shot Task Accuracy of Various Quantization Algorithms on LLaMA-2 13B.

| LLaMA-2 13B seqlen=4096 | bit | W2↓ | AC | AE | HE | QA | WI | Avg↑ |
|---|---|---|---|---|---|---|---|---|
| FP16 | 16 | 4.57 | 48.2 | 79.5 | 60.1 | 79.1 | 72.2 | 67.82 |
| GPTQ | 2 | 43.84 | 23.3 | 43.3 | 36 | 61.3 | 54.7 | 43.72 |
| GPTVQ | 2.25 | 5.72 | 38.7 | 73.6 | 51.6 | 75.4 | 68.5 | 61.56 |
| DB-LLM | 2.01 | 6.19 | 38.14 | 51.64 | 68.04 | 75.14 | 64.09 | 59.41 |
| AQLM | 2.18 | 5.41 | 39.42 | 69.15 | 54.68 | 76.22 | 68.43 | 61.58 |
| VPTQ | 2.02 | 5.32 | 40.02 | 71.55 | 56.18 | 77.26 | 66.85 | 62.37 |
| QuIP# | 2 | 5.35 | 39.5 | 69.3 | 56.01 | 77.3 | 67.7 | 61.96 |
| RSAVQ(ours) | 2 | 5.29 | 41.4 | 72.5 | 56.3 | 75.1 | 68.9 | 62.84 |
| GPTQ | 3 | 5.85 | 38.48 | 65.66 | 53.47 | 76.5 | 63.93 | 59.61 |
| GPTVQ | 3.125 | 4.8 | 44.45 | 77.23 | 58.18 | 77.8 | 71.98 | 59.63 |
| AQLM | 3.03 | 4.82 | 42.58 | 70.88 | 58.3 | 77.26 | 68.43 | 63.49 |
| VPTQ | 3.03 | 4.79 | 42.32 | 73.99 | 58.42 | 77.64 | 68.67 | 64.21 |
| QuIP# | 3 | 4.78 | 44 | 72.5 | – | 78.4 | 69.1 | – |
| RSAVQ(ours) | 3.01 | 4.74 | 44.9 | 77.7 | 57.5 | 78.1 | 72.4 | 66.12 |
| GPTQ | 4 | 4.78 | 42.49 | 70.45 | 58.67 | 77.75 | 70.01 | 63.87 |
| GPTVQ | 4.125 | 5.27 | 42.83 | 75.17 | 56.41 | 77.37 | 69.61 | 64.28 |
| AQLM | 3.94 | 4.65 | 44.8 | 73.32 | 59.27 | 78.35 | 69.85 | 65.12 |
| VPTQ | 4.02 | 4.64 | 44.37 | 73.19 | 59.37 | 77.75 | 69.77 | 64.89 |
| QuIP# | 4 | 4.63 | 45.5 | 73.9 | – | 78.9 | 69.9 | – |
| RSAVQ(ours) | 4.01 | 4.72 | 46.5 | 78.8 | 59 | 78.3 | 71.5 | 66.82 |

Table 11: Perplexity on Wikitext-2(Sequence Length=4096) and Zero-Shot Task Accuracy of Various Quantization Algorithms on LLaMA-2 70B.

| LLaMA-2 70B seqlen=4096 | bit | W2↓ | AC | AE | HE | QA | WI | Avg↑ |
|---|---|---|---|---|---|---|---|---|
| FP16 | 16 | 3.12 | 51.11 | 77.74 | 63.97 | 81.12 | 77.11 | 70.21 |
| GPTQ | 2 | NaN | 35.8 | 67 | 51.8 | 74.6 | 66.7 | 59.18 |
| GPTVQ | 2.25 | 4.25 | 49.4 | 80.47 | 58.26 | 79.4 | 75.2 | 68.55 |
| DB-LLM | 2.01 | 4.64 | 44.45 | 55.93 | 76.16 | 79.27 | 73.32 | 65.83 |
| AQLM | 2.07 | 3.94 | 47.93 | 77.68 | 61.79 | 80.43 | 75.93 | 68.75 |
| VPTQ | 2.07 | 3.93 | 47.7 | 77.1 | 62.98 | 80.3 | 74.98 | 68.61 |
| QuIP# | 2 | 3.91 | 48.7 | 77.3 | 62.49 | 80.3 | 75.9 | 68.94 |
| RSAVQ(ours) | 2 | 3.55 | 48.03 | 77.56 | 62.43 | 80.19 | 77.03 | 69.05 |
| GPTQ | 3 | 4.4 | 44.11 | 72.73 | 60 | 78.4 | 71.82 | 65.41 |
| AQLM | 3.01 | 3.36 | 50 | 77.61 | 63.23 | 81.28 | 77.19 | 69.86 |
| VPTQ | 3.01 | 3.34 | 48.89 | 77.06 | 63.52 | 80.9 | 77.51 | 69.58 |
| QuIP# | 3 | 3.35 | 50.9 | 77.7 | – | 81.4 | 76.4 | – |
| RSAVQ(ours) | 3 | 3.25 | 50.8 | 78.5 | 63.7 | 81.6 | 77.5 | 70.42 |
| GPTQ | 4 | 3.35 | 49.15 | 76.81 | 63.47 | 81.23 | 75.61 | 69.25 |
| AQLM | 4.14 | 3.19 | 50.68 | 77.31 | 63.69 | 81.5 | 76.48 | 69.93 |
| VPTQ | 4.01 | 3.19 | 49.57 | 78.16 | 63.71 | 81.18 | 76.4 | 69.8 |
| QuIP# | 4 | 3.18 | 50.6 | 78.1 | – | 81.4 | 77.1 | – |
| RSAVQ(ours) | 4.01 | 3.11 | 50.9 | 78.7 | 64 | 81.2 | 76.9 | 70.34 |

Table 12: Perplexity on Wikitext-2(Sequence Length=2048) and Zero-Shot Task Accuracy of Various Quantization Algorithms on LLaMA-3 8B.

| LLaMA-3 8B seqlen=2048 | bit | W2↓ | AC | AE | HE | QA | WI | Avg↑ |
|---|---|---|---|---|---|---|---|---|
| FP16 | 16 | 6.14 | 50.3 | 80.1 | 60.2 | 79.6 | 73.1 | 68.66 |
| GPTQ | 2 | 210 | 19.9 | 28.8 | 27.7 | 53.9 | 50.5 | 36.16 |
| DB-LLM | 2 | 13.6 | 28.2 | 59.1 | 42.1 | 68.9 | 60.4 | 51.74 |
| QuIP | 2 | 85.1 | 21.3 | 29 | 29.2 | 52.9 | 51.7 | 36.81 |
| QuIP# | 2 | 9.11 | 39.2 | 72.9 | – | 75.6 | 68.2 | – |
| VPTQ | 2.08 | 9.29 | 36.9 | 71 | 52.2 | 75.1 | 65.9 | 60.22 |
| RSAVQ(ours) | 2 | 8.79 | 40.9 | 74 | 51.9 | 75.7 | 66.1 | 61.72 |
| GPTQ | 3 | 8.2 | 37.7 | 70.5 | 54.3 | 74.9 | 71.1 | 61.7 |
| QuIP | 3 | 7.5 | 41 | 72.9 | 55.4 | 76.8 | 72.5 | 63.72 |
| QuIP# | 3 | 6.77 | 46.4 | 77.4 | – | 77.9 | 72.9 | – |
| VPTQ | 3.03 | 6.97 | 45.8 | 77.5 | 58.4 | 78.2 | 73.4 | 66.66 |
| RSAVQ(ours) | 3.01 | 6.34 | 46 | 76.1 | 57.7 | 78.2 | 73.9 | 66.38 |
| GPTQ | 4 | 6.5 | 47.7 | 78.8 | 59 | 78.4 | 72.6 | 67.3 |
| QuIP | 4 | 6.5 | 47.4 | 78.2 | 58.6 | 78.2 | 73.2 | 67.12 |
| QuIP# | 4 | 6.34 | 50.2 | 80.1 | – | 79.7 | 72.9 | – |
| VPTQ | 4.03 | 6.42 | 49.1 | 78.8 | 59.3 | 78.7 | 74.8 | 68.14 |
| RSAVQ(ours) | 4.01 | 6.31 | 48.5 | 79.7 | 59.8 | 78.9 | 75.2 | 68.42 |

Table 13: Perplexity on Wikitext-2(Sequence Length=2048) and Zero-Shot Task Accuracy of Various Quantization Algorithms on LLaMA-3 70B.

| LLaMA-3 70B seqlen=2048 | bit | W2↓ | AC | AE | HE | QA | WI | Avg↑ |
|---|---|---|---|---|---|---|---|---|
| FP16 | 16 | 2.9 | 60.1 | 87 | 66.3 | 82.4 | 80.8 | 75.32 |
| GPTQ | 2 | 11.9 | 24.6 | 38.9 | 41 | 62.7 | 59.9 | 45.42 |
| QuIP | 2 | 13 | 26.5 | 48.9 | 40.9 | 65.3 | 61.7 | 48.66 |
| QuIP# | 2 | 5.6 | 18.3 | 32.2 | – | 54.7 | 68.9 | – |
| VPTQ | 2.07 | 5.66 | 54.2 | 83.6 | 61.8 | 80.1 | 74 | 70.74 |
| RSAVQ(ours) | 2 | 5.6 | 54.4 | 83.1 | 61.7 | 80.2 | 77.1 | 71.3 |
| GPTQ | 3 | 5.2 | 52.1 | 79.6 | 63.5 | 80.6 | 77.1 | 70.58 |
| QuIP | 3 | 4.7 | 54.9 | 83.3 | 63.9 | 82.3 | 78.4 | 72.56 |
| QuIP# | 3 | 3.8 | 31.1 | 36.6 | – | 58.8 | 76.4 | – |
| VPTQ | 3.01 | 3.81 | 57.3 | 84.7 | 65.5 | 81.7 | 79.2 | 73.68 |
| RSAVQ(ours) | 3 | 3.69 | 58.1 | 85.2 | 67 | 81.1 | 79.9 | 74.26 |
| GPTQ | 4 | 3.3 | 58.4 | 86.3 | 66.1 | 82.9 | 80.7 | 74.88 |
| QuIP | 4 | 3.4 | 58.7 | 86 | 65.7 | 82.5 | 79.7 | 74.52 |
| QuIP# | 4 | 3.21 | 35 | 67.3 | – | 71.9 | 76.7 | – |
| VPTQ | 4.05 | 3.15 | 59 | 86.1 | 66.2 | 82.4 | 79.8 | 74.7 |
| RSAVQ(ours) | 4.01 | 3.11 | 59.2 | 86.4 | 66.1 | 83.4 | 80.4 | 75.1 |

