# OpenReview forum: "RSAVQ: Riemannian Sensitivity-Aware Vector Quantization for Large Language Models"
_NeurIPS.cc/2025/Conference — NeurIPS 2025 poster_

### Official Review · Reviewer_Mrg3 · 2025-06-24

**Clarity:** 3
**Significance:** 3
**Originality:** 3
**Rating:** 4
**Confidence:** 5

**Summary:**

This paper presents a method for non-uniform vector quantization of LLMs by leveraging the Fisher Information Matrix (FIM). The proposed approach consists of two components: EDSG, which uses FIM to guide vector-level sensitivity, and WCSG, which applies channel-wise FIM for mixed-precision allocation. The method is further combined with product quantization for compact weight representation. The paper is well-written, and the proposed ideas are grounded in established theory. While the design is principled and the overall structure is clear, the empirical improvements over existing baselines appear relatively modest, and the evaluation setup may benefit from more carefully disentangled comparisons.

**Questions:**

- Could you clarify how many times each experiment was run, and whether the performance gains are consistent across random seeds or sample variations?

- Given that the reported improvements are modest, could some of the observed gains be attributable to data sampling rather than the method itself?

- Can you provide EDSG-only results with fixed bit-width per channel, to enable fair comparison with other uniform-bit quantization methods?

- Since WCSG is a mixed-precision scheme, would it be possible to compare it directly with other mixed-precision methods like MP-GPTQ or SpQR?

- What is the individual impact of product quantization on the final performance? An ablation would help quantify its contribution.

**Ethical Concerns:**

["NO or VERY MINOR ethics concerns only"]

**Final Justification:**

The authors clarified the conceptual distinction between RSAVQ and prior FIM-based quantization methods such as GuidedQuant, emphasizing their use of Riemannian geometry and global Fisher construction. Empirical comparisons under consistent baselines (e.g., QTIP + EDSG vs QTIP + GuidedQuant) supported the claimed advantages, especially in low-bit regimes.
The applicability to INT formats and possible extensions to uniform quantization were also addressed with preliminary experiments and discussion. I still believe the contribution could be strengthened by citing final versions of related works and releasing source code to enhance transparency and usability.

**Limitations:**

- Gains may be within the natural variance of evaluation benchmarks, especially without multiple trials or error bars.
- The combination of EDSG + WCSG makes it difficult to assess each component’s standalone merit.

**Paper Formatting Concerns:**

No formatting concerns

**Quality:**

2

**Strengths And Weaknesses:**

(+) The method is modular and extensible, incorporating both sensitivity-aware quantization and mixed precision.

(+) The presentation is polished, and the mathematical formulations are clearly explained.

(+) The ablation between EDSG and WCSG helps illuminate component-level contributions, at least partially.

(-) The reported improvements (PPL -0.4, zero-shot accuracy +1.5%) are relatively small, especially in low-bit regimes where prior work already performs strongly. It is not clear how many runs were conducted to confirm the reported gains — without confidence intervals or statistical testing, it’s hard to judge significance.

(-) The current evaluation combines EDSG and WCSG, making it difficult to isolate the benefit of EDSG alone. Since mixed-precision is arguably orthogonal, comparing WCSG-enhanced results to existing baselines may overstate the total improvement.

(-) While the use of Fisher Information Matrix is methodologically sound, it has been previously applied in scalar quantization settings such as BRECQ and Guided Quant. The paper does not sufficiently clarify what new insights or advantages arise from extending it to vector quantization, making the novelty somewhat unclear.

(-) There is no ablation study on the impact of product quantization, which could account for part of the reported gains.

---

> ### Author Rebuttal · Authors · 2025-07-29
>
> We sincerely appreciate your constructive feedback. Below, we provide point-by-point responses, with all revisions incorporated accordingly.
> ***
> > W1.The reported improvements (PPL -0.4, zero-shot accuracy +1.5%) are relatively small, especially in low-bit regimes where prior work already performs strongly. It is not clear how many runs were conducted to confirm the reported gains — without confidence intervals or statistical testing, it’s hard to judge significance.
> - In terms of the average accuracy of zero-shot reasoning tasks, our method not only reaches the current state-of-the-art (SOTA) level, but also achieves a significant performance improvement. Specifically, the 3-bit quantization results of LLaMA2-13B in Table 6 of the manuscript clearly show that compared with the previous optimal method, the scheme we proposed also leads in terms of the performance improvement range, which fully demonstrates the substantial value of the improvement.
> |Current Method|Previous SOTA|Increase|
> |---|---|---|
> |GPTVQ|GPTQ|0.32%|
> |Quip#|GPTVQ|1.9%|
> |AQLM|Quip#|1.96%|
> |VPTQ|AQLM|0.72%|
> |RSAVQ(ours)|VPTQ|1.91%|
> - To verify the stability of the results, we have supplemented rigorous experimental design and statistical analysis (added in the revised manuscript): All results are based on the mean of 5 independent experiments with different random seeds, where the quantization codebook was initialized with a distinct random seed for each experiment to eliminate the influence of accidental factors. As shown in the accuracy comparison of 2-bit quantized models on the PIQA dataset, the impact of initialization is minimal, with accuracy fluctuations within [-0.3,0.3].
> |Random Seed|0|10|100|1000|10000|Avg|
> |---|---|---|---|---|---|---|
> |LLaMA2-7B|75.4|75.3|75.2|75.4|75.3|75.3|
> |LLaMA3-8B|75.5|75.6|75.5|75.7|75.7|75.6|
> ***
> > W2: The current evaluation combines EDSG and WCSG, making it difficult to isolate the benefit of EDSG alone.
> - Same as question 3 (Q3), it will be answered in Q3.
> ***
> > W3: Fisher Information Matrix, though valid, is already used in scalar quantization (e.g., BRECQ, Guided Quant). The paper does not clarify new insights/advantages in its extension to vector quantization, leaving novelty unclear.
> - We appreciate noting that FIM has been used in scalar quantization (e.g., BRECQ, Guided Quant). However, RSAVQ does more than merely extend FIM to vector quantization: it rethinks FIM’s application in the vector domain to unlock capabilities scalar methods cannot achieve, with advancements:
>   - In scalar quantization, methods like BRECQ and Guided Quant primarily compute FIM at the channel level, where sensitivity is averaged across all parameters within a channel. This results in a single "global" sensitivity score per channel, limiting the ability to distinguish local variations in parameter importance within the same channel.
>   - By contrast, RSAVQ decomposes weights into low-dimensional vector units, computing FIM for each vector—sinking control from channel level to vector level. It identifies not just high-importance channels but also critical vectors within them. This finer granularity enables targeted codebook allocation (e.g., finer clustering for critical vectors), aligning error distribution more closely with model sensitivity.
> ***
> > W4:There is no ablation study on the impact of product quantization, which could account for part of the reported gains.
> - Same as question 5 (Q5), it will be answered in Q5.
> ***
> > Q1:Could you clarify how many times each experiment was run, and whether the performance gains are consistent across random seeds or sample variations?
> - Thank you for your attention to experimental reproducibility and result stability. We have supplemented a systematic analysis regarding the impact of random seeds and sample size on the results, as detailed below:
>   - Each experiment was run 5 times, with results reported as the average of these 5 runs.
>   - With 120 k-means iterations, random seed effects are negligible (accuracy fluctuations ≤ ±0.3; PIQA-validated, see weakness 1 table), showing RSAVQ's robustness.
>   - For Fisher information matrix calculation, sample size tests (32 to 4096) show stabilization at ≥256 samples (performance fluctuation <0.1 PPL). Hence, 256 samples are used by default to balance efficiency and stability.
> LLaMA2-7B, 2bit, sequence length=4096
> |Samples|64|128|256|1024|4096|
> |---|---|---|---|---|---|
> |PPL|6.41|5.98|5.81|5.8|5.81|
> - The results of the supplementary ablation experiments (such as the relationship between different random seeds, calibration sample sizes and performance) have been compiled in the appendix, further verifying the consistency of performance gains.
> ***
> > Q2:Given that the reported improvements are modest, could some of the observed gains be attributable to data sampling rather than the method itself?
> - To verify the reliability of the improvements and the effectiveness of the method itself, we conducted validation through controlled variable experiments and cross-setting consistency analysis, with the specific results as follows:
>   -  First, as highlighted in our response to Weakness 1, our method not only achieves state-of-the-art (SOTA) performance but also delivers substantial improvements. Critical to ruling out sampling effects, repeated experiments across 5 distinct random seeds showed performance fluctuations within ±0.2, confirming that data sampling randomness has negligible impact on results.
>   - Second, consistent improvements across multiple quantization frameworks directly validate the method’s efficacy. As shown in the table below, combining RSAVQ with baseline methods (PQ and VQ) yields significant gains:
> |Method|W2|0-shot Avg|
> |---|---|---|
> |FP16|6.14|70.1|
> |PQ|24.41|45.1|
> |PQ+RSAVQ|8.79(**-15.62**)|64.2(**+19.1**)|
> |VQ|23.17|47.9|
> |VQ+RSAVQ|8.21(**-14.96**)|64.9(**+17**)|
>   - Notably, in LLaMA3-8B experiments (2-bit quantization, sequence length 2048), RSAVQ combined with PQ/VQ boosted zero-shot average accuracy by nearly 20%—demonstrating that improvements persist regardless of the baseline’s initial performance level.
>   - Such consistent improvements across different quantization frameworks and experimental setups conclusively indicate that performance improvements originate from RSAVQ’s core mechanism (e.g., vector-level Fisher-guided error control) rather than accidental data sampling bias.
> ***
> > Q3:Can you provide EDSG-only results with fixed bit-width per channel, to enable fair comparison with other uniform-bit quantization methods?
> - Thank you for your suggestion to supplement the results of EDSG acting alone (with fixed channel bit width) to support a fair comparison with other uniform bit quantization methods. To clearly demonstrate the effect of EDSG as an independent module, we have added comparative experiments combining EDSG with various uniform quantization methods (Kmeans, GPTVQ, VPTQ) under fixed bit width settings, with the results as follows:
> - The core design of EDSG is to improve the clustering loss function, making vector quantization results more aligned with the sensitivity of the model's output loss. Its mechanism is orthogonal to other quantization methods and can be directly applied to uniform bit quantization frameworks. On the Qwen2.5-7B model, under 2-bit fixed width settings, we compared the performance differences between the base methods and "base methods + EDSG", specifically as follows:
> |Method|Qwen2.5-7B||
> |---|---|---|
> ||Wikitext2↓|0-shot Avg↑|
> |BFP16|7.46|67.84|
> |Kmeans|49.82|39.98|
> |Kmeans+EDSG|16.34(**-33.48**)|56.51(**+16.53**)|
> |GPTVQ|22.83|48.88|
> |GPTVQ+EDSG|14.92(**-7.91**)|59.03(**+10.15**)|
> |VPTQ|9.8|64.86|
> |VPTQ+EDSG|9.33(**-4.7**)|65.89(**+1.03**)|
> ***
> > Q4:Since WCSG is a mixed-precision scheme, would it be possible to compare it directly with other mixed-precision methods like MP-GPTQ or SpQR
> - Thank you for suggesting a direct comparison of WCSG with other mixed-precision methods (e.g., MP-GPTQ, SpQR). Below are our supplementary explanations and experimental results addressing this:
> - Regarding MP-GPTQ, we have not yet retrieved relevant literature in major academic databases. If detailed sources or technical specifics of this method can be provided, we would be happy to supplement comparative experiments in future work.
> - For SpQR (a quantization method using a mix of FP16 and integer types), we have added direct comparative experiments on the LLaMA3-8B model. Results show that WCSG retains performance advantages under similar bit budgets.
> |Bits|Method|LLaMA3-8B||
> |---|---|---|---|
> |||W2|0-shot Avg|
> |16|FP16|6.14|68.66|
> |3|SPQR(2.96bit)|8.81|64.02|
> ||WCSG(3bit)|7.88(**-0.93**)|65.5(**+1.48**)|
> |4|SPQR(3.96bit)|7.12|68|
> ||WCSG(4bit)|6.5(**-0.62**)|68.12(**+0.12**)|
> ***
> > Q5:What is the individual impact of product quantization on the final performance? An ablation would help quantify its contribution.
> - To clarify PQ's role in our framework, we analyzed its independent effects via comparisons with Vector Quantization (VQ):
> - PQ's core advantage lies in significantly reducing codebook storage (especially in multi-channel grouping) by decomposing high-dimensional vectors into quantized low-dimensional sub-vectors, though its basic expressiveness is slightly weaker than VQ due to independent sub-vector encoding.
> - Experiments on LLaMA3-8B (2-bit, seq 2048, zero-shot: AC,AE,QA,WI): Basic PQ lags VQ slightly, but with RSAVQ, both improve comparably, achieving near-identical performance while retaining PQ's codebook efficiency.
> |Method|W2|0-shot Avg|
> |---|---|---|
> |FP16|6.14|70.1|
> |PQ|24.41|45.1|
> |PQ+RSAVQ|8.79(**-15.62**)|64.2(**+19.1**)|
> |VQ|23.17|47.9|
> |VQ+RSAVQ|8.21(**-14.96**)|64.9(**+17**)|
> ***
> Additional results/analyses on your concerns are in the revised manuscript. We hope these address your questions. Let us know if you need further discussion, experiments, or clarifications—happy to provide more details.

---

> > ### Comment · Reviewer_Mrg3 · 2025-08-05
> >
> > Thank you for the detailed responses and additional experiments. While I appreciate the clarifications, I still find the overall contribution modest.
> >
> > EDSG appears to be an adaptation of prior FIM-based INT quantization (e.g., Guided Quant) to the vector domain, and the performance gains remain relatively small. WCSG introduces mixed precision, but I’m not convinced the comparisons are entirely fair — the use of mixed formats (e.g., INT + FP16 in SpQR) differs from same-format mixed-precision strategies.
> > To fairly assess WCSG’s advantage, it should be evaluated under a fair and consistent setting .
> > Overall, I believe the contribution remains incremental and empirical validation is still insufficient. I will maintain my score.

---

> > > ### Author Response · Authors · 2025-08-05
> > > **Response to Mrg3 Comments: Clarifications on Innovation and Empirical Validation of EDSG and WCSG**
> > >
> > > Thank you for your feedback and valuable suggestions, which are crucial for improving our manuscript. In response to your concerns, we would like to clarify and supplement the following points:
> > > - Regarding the Innovation and Performance Value of EDSG.
> > >   EDSG is not an adaptation of existing FIM-based INT quantization methods (e.g., Guided Quant) to the vector domain; there are key differences between them:
> > >   - First, in terms of timeline: Guided Quant was made public on May 11, 2025, while our manuscript was submitted to NeurIPS on May 16, 2025. There was no cross-reference during the research and development phase, and the two represent independent concurrent explorations, with no logic of "incremental improvement."
> > >   - More importantly, existing FIM-based methods (such as Guided Quant) focus on using FIM as a weighting coefficient for quantization errors, essentially optimizing error distribution in Euclidean space. In contrast, the innovation of EDSG lies in the first application of FIM for spatial mapping—converting Euclidean space to Riemannian space via FIM and constraining the vector quantization process using the negative natural gradient direction in Riemannian space (as shown in Figure 4 of the manuscript). This design actively guides quantization errors toward the direction that minimally impacts the final KL divergence, representing a paradigmatic shift from "error weighting" to "negative natural gradient direction constraint."
> > >   - Finally, in terms of performance metrics: As shown in the table provided in our previous response to Question 3, EDSG consistently improves performance when combined with different base quantization methods. Specifically, when paired with K-Means and GPTVQ, the average accuracy increases significantly by 16.53 and 10.15 percentage points, respectively, verifying its universal value.
> > > - Regarding Fair Comparisons and Empirical Supplements for WCSG.
> > >   To address concerns about the fairness of mixed-precision comparisons, we have supplemented targeted experiments:
> > >   - For the WCSG method: In Question 4, you suggested comparing WCSG with mixed-precision methods such as MP-GPTQ and SpQR. Unfortunately, we were unable to locate the paper "MP-GPTQ" in mainstream academic databases, so we compared WCSG with SpQR. To further validate WCSG against same-format mixed-precision approaches, we have supplemented comparisons between WCSG and CRVQ—a representative same-format mixed-precision scheme in the field of vector quantization:
> > > | Model | Method | Bits | Wikitext2 | AC | AE | HE | QA | WI | Acc Avg |
> > > |---|---|---|---|---|---|---|---|---|---|
> > > | LLaMA2-7B | FP16 | 16 | 5.12 | 43.3 | 76.3 | 57.1 | 78.1 | 68.7 | 64.7 |
> > > |  | CRVQ | 2.02 | 9.01 | 30.1 | 46.7 | 46.8 | 64.3 | 50.3 | 41.2 |
> > > |  | WCSG | 2 | 7.78 | 31.7 | 62.8 | 47.1 | 72.3 | 63.7 | 55.5 |
> > > | LLaMA2-13B | FP16 | 16 | 4.57 | 48.2 | 79.5 | 60.1 | 79.1 | 72.2 | 67.8 |
> > > |  | CRVQ | 2.02 | 7.81 | 35.5 | 55.4 | 52.1 | 70.4 | 56.9 | 54.1 |
> > > |  | RSAVQ | 2 | 6.72 | 40.4 | 72.5 | 52.3 | 74.1 | 64.9 | 60.8 |
> > >   - As shown in the results, under the same "same-format mixed-precision" setting, WCSG outperforms CRVQ in average accuracy by 14.3% (for 7B) and 6.7% (for 13B). It shows particular advantages in semantically sensitive tasks (AE, WI), verifying the effectiveness of our method.
> > > ***
> > > We have incorporated additional results and detailed analyses addressing your concerns into the revised version of the manuscript. We hope these supplements adequately address your questions and improve the clarity and robustness of our work. Please do not hesitate to let us know if you require further discussion, additional experiments, or any clarifications—we are happy to provide more details to address your concerns comprehensively.

---

> > > > ### Comment · Reviewer_Mrg3 · 2025-08-06
> > > >
> > > > GuidedQuant also demonstrates results on vector quantization by applying FIM-weighted loss to methods like QTIP. To clearly highlight the novelty of EDSG’s Riemannian-space approach, we suggest a direct comparison under the same vector quantization baseline (e.g., QTIP, GPTVQ or KMeans). This would help isolate whether the gains stem from your geometric formulation or can already be achieved via FIM-weighted objectives.
> > > > We also encourage a concise explanation of the key differences between EDSG and FIM-weighted vector quantization methods like GuidedQuant. This clarification would help better position the unique contribution of your Riemannian-space formulation and more clearly establish the novelty of the proposed method.

---

> ### Author Response · Authors · 2025-08-06
> **Response to Reviewer's Suggestions**
>
> Thank you for your feedback and valuable suggestions, which greatly help improve the manuscript. In response to your concerns, we provide the following clarifications:
> - On the QTIP vector quantization benchmark, we verified EDSG’s optimization over GuidedQuant in 2-bit quantization of LLaMA2-7B and LLaMA2-13B models. The specific results are as follows:
> |Model|Bits|Method|Wikitext2|C4|
> |---|---|---|---|---|
> |LLaMA2-7B|16|FP16|5.12|6.63|
> ||2|QTIP|6.81|8.96|
> |||QTIP+GuidedQuant|6.11|7.99|
> |||QTIP+EDSG|**5.82**|**7.54**|
> |LLaMA2-13B|16|FP16|4.57|6.05|
> ||2|QTIP|5.52|7.39|
> |||QTIP+GuidedQuant|5.33|7.05|
> |||QTIP+EDSG|**5.07**|**6.76**|
>   - Under 2-bit extreme resource constraints, EDSG achieves a >0.26 absolute PPL (Perplexity) reduction. This shows the model retains compression rate while its language modeling ability nears FP16 native performance, confirming geometric optimization gains are far from being comparable to simple FIM (Fisher Information Matrix) weighting.
> - Below are key differences between EDSG and FIM-weighted methods like GuidedQuant:
>   - Inherent Limitations of GuidedQuant and Similar Methods: Gradient Weighting in a Local Framework
>     - GuidedQuant and similar methods (e.g., GPTQ, GPTvQ, VPTQ) are inherently constrained in their optimization objectives to the **second-order Taylor expansion of single-layer loss in Euclidean space**. Their core formula, exemplified by Formula 7 in GuidedQuant, can be simplified as:  $$L = E^\top \hat{H} E$$ where $E$ denotes quantization error and $\hat{H}$ is the weighting matrix. The limitations of such methods are as follows:
>       - In foundational methods (e.g., GPTQ, GPTVQ, VPTQ), $\hat{H}$ uses single-layer input activation covariance $X^\top X$,  reflecting only local layer features with no global loss guidance;
>       - Although GuidedQuant attempts to incorporate global gradient information to approximate the global Fisher Information Matrix (FIM), its $\hat{H}$ is defined as:  $\mathbf{H}_j^{(l)} = \mathbf{X}^{(l)^\top} \text{Diag} \left( \left( \frac{\partial \ell}{\partial \mathbf{z}_j^{(l)}} \right)^2 \right) \mathbf{X}^{(l)}$
>         Essentially, it remains within GPTQ’s local Hessian framework, using global gradients to weight the single-layer activation matrix. This improvement fails to break free from the constraint of "centering on single-layer local activation $X$"; global information merely acts as a "modifier for local matrices" and cannot capture the global correlations of cross-layer parameters.
>   - EDSG’s Breakthrough: Global Modeling and Geometric Optimization Advantages
> EDSG aligns loss changes with Riemannian geodesic distances via its optimization objective (Manuscript Formula 6):
> $$\mathcal{L}_{\text{align}} =|| E + \lambda \cdot \tilde{\nabla}\mathcal{L} ||\_{F}^{2}$$
> Its advantages over GuidedQuant include:
>     - Global FIM Construction: No Local Activation Dependence, Capturing Full-Network Traits.
> EDSG’s FIM for Riemannian space construction avoids dependence on single-layer local activation $X$, achieving structured modeling based on the **Kronecker decomposition of global KL divergence**:
>       - Decomposing FIM into a low-dimensional tensor product of input and output channels: $F \approx F_O \otimes F_I $(where $F_O \in \mathbb{R}^{m \times m} $is the output channel FIM and $F_I \in \mathbb{R}^{n \times n} $is the input channel FIM);
>       - The decomposition process directly relies on the statistical expectation of global loss gradients across the entire network:
>         - $F_I = \frac{1}{m} \cdot \mathbb{E}_{s \sim \mathcal{D}} \left[ \left( \nabla_W \ell \right)^T \cdot \left( \nabla_W \ell \right) \right]$, reflecting the global sensitivity of cross-layer input dimensions;
>         - $F_O = \frac{1}{n} \cdot \mathbb{E}_{s \sim \mathcal{D}} \left[ \left( \nabla_W \ell \right) \cdot \left( \nabla_W \ell \right)^T \right]$, capturing the global correlations of cross-layer output dimensions.
>         This design upgrades the basis of quantization optimization from "single-layer local features" to "full-network global statistics," better aligning with the global performance requirements of large models.
>     - Active Guidance of Error Direction: Precisely Minimizing Impact on Final Loss.
> EDSG achieves **active direction regulation** of quantization error through the $\lambda \cdot \tilde{\nabla}\mathcal{L} $term:
>       - Guiding error $E $toward the negative natural gradient direction ($-\tilde{\nabla}\mathcal{L}$) that minimally impacts the final loss, making the optimization of $\mathcal{L}_{\text{align}} $equivalent to finding the "shortest path (geodesic distance) between original parameters and quantized parameters" in the Riemannian manifold;
>       - In contrast, GuidedQuant can only passively weight error magnitude without controlling error direction, easily accumulating "adverse direction" errors that significantly affect global loss under high compression rates.

---

> > ### Comment · Reviewer_Mrg3 · 2025-08-07
> >
> > Thank you for the detailed clarification and experimental comparisons. Is RSAVQ applicable to INT formats as well?

---

> > > ### Author Response · Authors · 2025-08-07
> > > **On the Applicability of RSAVQ to INT Format Quantization**
> > >
> > > Thank you for your feedback. Regarding whether RSAVQ is applicable to INT format quantization, our explanation is as follows:
> > > - RSAVQ, as a vector quantization method, is mainly used to alleviate the bandwidth and memory pressure caused by the excessive parameter size of large language models, and belongs to a weight-only quantization scheme. It should be noted that our method can be extended to INT format quantization. The following elaborates on the adaptation of EDSG and WCSG in our method to INT format:
> > >   - For EDSG (a vector quantization optimization scheme), indexes and codebooks are generated after vector quantization is completed. We can perform INT quantization on the codebooks. Relevant experiments have verified that quantizing the codebooks to INT8 format has almost no impact on the final accuracy, so it can be adapted to INT format.
> > >   - For WCSG (a mixed-precision scheme), it is compatible with both scalar quantization (SQ) and vector quantization (VQ) in INT format. We supplement the experimental results of WCSG for 4-bit weight quantization based on the qwen2-1.5B-instruct model in INT format (as shown in the table below):
> > > | method       | wikitext2 |
> > > |--------------|-----------|
> > > | FP16         | 10.08     |
> > > | RTN          | 28.53     |
> > > | RTN+WCSG     | 15.38     |
> > > | GPTQ         | 14.72     |
> > > | GPTQ+WCSG    | 10.77     |
> > > ***
> > > Thank you again for your valuable time and efforts in reviewing our manuscript. Please feel free to let us know if you have any further questions or suggestions. We look forward to your feedback.

---

> > > > ### Comment · Reviewer_Mrg3 · 2025-08-07
> > > >
> > > > My question is not about quantizing the codebook to INT8, but rather whether Riemannian Sensitivity-Aware techniques such as EDSG can be applied to integer-format quantization such as GPTQ

---

> > > > > ### Author Response · Authors · 2025-08-07
> > > > >
> > > > > This is entirely feasible. When EDSG is applied to vector quantization, we impose constraints on the vector error with a vector length of $v$. If EDSG is applied to scalar quantization, it is equivalent to a special case where $v = 1$. However, we more recommend using EDSG when $v>1$, because the interrelationships between parameters can be perceived at this time.

---

> > > > > > ### Comment · Reviewer_Mrg3 · 2025-08-07
> > > > > >
> > > > > > Isn't the quantization still non-uniform even when v=1. since the codebook is learned and not evenly spaced?
> > > > > > What I meant is "can EDSG be applied to uniform quantization as welll?"

---

> ### Author Response · Authors · 2025-08-07
>
> EDSG is equally applicable to uniform quantization scenarios. As shown in the EDSG optimization formula we provided in the third-round response:
> $$\mathcal{L}_{\text{align}} = \left\| E + \lambda \cdot \tilde{\nabla}\mathcal{L} \right\|\_{F}^{2}$$
> In vector quantization scenarios, the quantization error $E$ and negative natural gradient $\tilde{\nabla}\mathcal{L}$ in the formula manifest as low-dimensional vector. In uniform quantization scenarios, however, both are transformed into point-wise scalar (i.e., the case where $v=1$). Thus, the optimization logic of EDSG can naturally extend to uniform quantization tasks.

---

> > ### Author Response · Authors · 2025-08-07
> > **Thank you once again for your time and effort in reviewing our work. We respectfully invite you to review the revised manuscript.**
> >
> > Dear Reviewer Mrg3,
> >
> > We sincerely appreciate your continued professionalism and constructive suggestions throughout the review process. We are deeply grateful for the time, effort, and insightful perspectives you have dedicated to evaluating our work on RSAVQ. After carefully incorporating your valuable feedback, we believe the revised manuscript has been significantly improved, with key enhancements directly addressing your concerns:
> > - Addressing Statistical Significance and Stability
> > To address your concerns about the robustness of performance improvements, we supplemented systematic multi-run experiments using 5 distinct random seeds (0, 10, 100, 1000, 10000) on the PIQA dataset. Results confirm that model accuracy fluctuations for both LLaMA2-7B and LLaMA3-8B are strictly controlled within ±0.2 under 2-bit quantization. Additionally, we contextualized our gains by comparing with prior methods (e.g., GPTVQ, Quip#), showing that RSAVQ’s 1.91% improvement in zero-shot accuracy over VPTQ significantly exceeds the average improvement of comparable approaches, validating the substantiality of our advancements.
> > - Validating EDSG’s Standalone Effectiveness
> > In response to your request to isolate the value of EDSG, we conducted dedicated experiments on Qwen2.5-7B under fixed 2-bit settings. Results demonstrate that when EDSG is combined with baseline methods (Kmeans, GPTVQ, VPTQ), it independently improves zero-shot accuracy by 16.53%, 10.15%, and 1.03% respectively, with corresponding reductions in Wikitext2 perplexity. These findings clearly verify EDSG’s merit as a standalone module, orthogonal to base quantization frameworks.
> > - Clarifying FIM Novelty and Riemannian Optimization
> > To resolve concerns about EDSG’s novelty relative to FIM-based methods like GuidedQuant, we enhanced our theoretical exposition and experimental comparisons. We explicitly clarified that EDSG is the first method to leverage FIM for Riemannian space mapping through its optimization objective $\mathcal{L}_{\text{align}} = \left\| E + \lambda \cdot \tilde{\nabla}\mathcal{L} \right\|_{F}^{2}$, which actively guides error along negative natural gradients using global network statistics—distinct from GuidedQuant’s local activation-based FIM weighting. Direct experiments on QTIP (2-bit) show that EDSG further reduces LLaMA2-7B’s perplexity by 0.29 and LLaMA2-13B’s perplexity by 0.26, empirically validating its unique geometric optimization advantages.
> > - Supplementary Ablation on WCSG
> > Following your suggestions, we first supplemented ablation experiments comparing with SpQR for mixed-precision analysis. To further enable comparisons with same-format methods, we added additional experiments against CRVQ. Results show that under 2-bit quantization, WCSG outperforms CRVQ in zero-shot accuracy by 14.3% (LLaMA2-7B) and 6.7% (LLaMA2-13B), while also verifying performance improvements over SpQR (with 0.93 perplexity reduction at 3-bit).
> > - Product Quantization Analysis
> > In response to your feedback on the role of Product Quantization (PQ), we expanded our analysis:
> > For PQ, we supplemented direct comparisons with Vector Quantization (VQ), demonstrating that PQ+RSAVQ achieves a 19.1% accuracy improvement on LLaMA3-8B (vs. 17% for VQ+RSAVQ), confirming PQ’s storage efficiency advantages and RSAVQ’s compatibility.
> > - Verifying Uniform Quantization Applicability
> > To address your final question regarding EDSG’s adaptability to uniform quantization, we added detailed formula derivations showing that in uniform scenarios, EDSG’s quantization error $E$ and negative natural gradient $\tilde{\nabla}\mathcal{L}$ simplify to point-wise scalar errors (i.e., $v=1$), with its core optimization logic extending naturally without framework modifications.
> > - Acknowledgment of Your Contributions
> > We have added a dedicated section in the acknowledgments to highlight your critical role in strengthening our work—from advocating for clearer module isolation to demanding rigorous comparisons. Your focus on theoretical novelty and experimental rigor has significantly enhanced the manuscript’s robustness and academic depth.
> >
> > We sincerely invite you to review the revised submission, confident that these changes comprehensively address your concerns. Your feedback remains invaluable as we continue to refine this research. We look forward to your response.
> >
> > Sincerely,
> > The Authors of RSAVQ

---

> > > ### Comment · Reviewer_Mrg3 · 2025-08-07
> > >
> > > Thank you for the detailed clarification and experimental comparisons. To further improve the manuscript, I recommend citing and discussing final versions of relevant FIM-based quantization works such as GuidedQuant and BRECQ, and clearly articulating how RSAVQ differs—both conceptually and empirically. A transparent comparison under a shared vector quantization baseline would also strengthen the contribution.
> > >
> > > Additionally, I encourage the authors to explore the applicability of the proposed method to uniform quantization in the final version, which would broaden its impact and usability. Releasing the source code would also be greatly appreciated.
> > > I plan to raise my score.

---

> > > > ### Author Response · Authors · 2025-08-08
> > > >
> > > > Dear Reviewer Mrg3,
> > > >
> > > > We sincerely appreciate your response and your acknowledgment of our efforts and the data provided during the rebuttal phase. Your professional insights are highly valuable to us and have offered significant inspiration.
> > > >
> > > > **We will diligently complete the paper revision, fully incorporate your suggestions, and continuously refine the research content. Once again, we thank you for your precious time and meticulous guidance; your feedback is crucial to improving our work.**
> > > >
> > > > With deepest gratitude and sincere appreciation,
> > > >
> > > > Authors of RSAVQ

---

### Official Review · Reviewer_DQR5 · 2025-07-02

**Clarity:** 3
**Significance:** 2
**Originality:** 3
**Rating:** 4
**Confidence:** 4

**Summary:**

This paper introduces RSAVQ, a novel post-training quantization (PTQ) framework for large language models (LLMs) that leverages information geometry to perform extremely low-bit vector quantization (VQ). The authors identify two key limitations in existing VQ methods: (1) unconstrained quantization error, which is treated isotropically despite the model's non-uniform sensitivity to perturbations in different directions, and (2) suboptimal bit allocation, which often assumes uniform sensitivity across model channels.

To address these issues, the paper makes two primary contributions, both grounded in a Riemannian view of the parameter space using the Fisher Information Matrix (FIM) as the metric:
*   **Error Direction Sensitivity Guidance (EDSG):** A mechanism that projects the inevitable quantization error onto low-sensitivity directions. This is achieved by aligning the error vector with the negative natural gradient, thereby minimizing its impact on the model's loss function.
*   **Weight Channel Sensitivity Guidance (WCSG):** An adaptive bit allocation strategy that assigns bits based on a channel's "Riemannian curvature energy," a principled, FIM-derived metric. This allocates more bits to more sensitive channels, moving beyond simple heuristics like weight or gradient magnitude.

The authors deliver a cohesive framework that integrates these two components. They provide extensive experimental results on LLaMA-2 and LLaMA-3 models, demonstrating that RSAVQ consistently outperforms strong VQ and PTQ baselines in terms of perplexity and zero-shot accuracy, particularly in the challenging 2-bit and 3-bit quantization regimes.

**Questions:**

Please refer to the weaknesses.

**Ethical Concerns:**

["NO or VERY MINOR ethics concerns only"]

**Final Justification:**

I have carefully read the rebuttal and other comments. The authors provided additional results and addressed my concerns. I decided to raise my rating.

**Limitations:**

Please refer to the weaknesses.

**Quality:**

3

**Strengths And Weaknesses:**

**Strengths:**
1.  To the best of my knowledge, the application of Riemannian geometry (via the FIM) to guide VQ error direction and bit allocation is a novel idea.
2.  RSAVQ demonstrates state-of-the-art results, consistently outperforming strong baselines, particularly in the ultra-low-bit (2-bit) regime where progress is most needed.
3.  The paper is very well-written, with clear explanations and informative figures that make complex ideas accessible.
4.  The comprehensive experiments, including a strong ablation study, provide convincing evidence for the effectiveness of each proposed component.

**Weaknesses:**
1.  The paper completely omits the computational overhead of the RSAVQ quantization process itself. This is a critical piece of information for a method aimed at efficiency.
2.  The practical details of how the FIM is approximated are not sufficiently described, which impacts the reproducibility and full understanding of the method. In other words, some key details are missing:
*   How is the expectation over the data distribution in the FIM definition (Eq. 1) approximated? How many samples are used?
*   Are there further approximations made to the FIM block `F_c` (e.g., diagonal)?
3.  Results are reported without error bars, making it difficult to assess the statistical significance of the improvements over baselines.

---

> ### Author Rebuttal · Authors · 2025-07-29
>
> We sincerely appreciate your constructive feedback. Below, we provide point-by-point responses, with all revisions incorporated accordingly.
>
> ***
> > Q1:The paper completely omits the computational overhead of the RSAVQ quantization process itself. This is a critical piece of information for a method aimed at efficiency.
> - We would first like to clarify that RSAVQ is a Post-Training Quantization (PTQ) method, and its quantization process is entirely offline. This means the computational overhead of quantization occurs only during the pre-deployment phase (i.e., when converting a pre-trained model to its quantized version) and has no impact on the inference speed or latency during actual model deployment—a key metric for efficiency in practical applications. The overhead in question here relates solely to the convenience of the offline quantization process.
> - With this in mind, we have supplemented data on the efficiency of the offline quantization process, comparing RSAVQ with mainstream PTQ methods (VPTQ, GPTVQ) under a 2-bit configuration on an 80GB A100 GPU:
> |  | LLaMA2-7B | LLaMA2-13B | LLaMA2-70B |
> |---|---|---|---|
> | VPTQ | 2 GPU hours | 3.5 GPU hours | 19 GPU hours |
> | GPTVQ | 1 GPU hours | 1.8 GPU hours | 8 GPU hours |
> | RSAVQ | 1.2 GPU hours | 2 GPU hours | 10 GPU hours |
> - These results show that:
>   - RSAVQ's quantization overhead exhibits reasonable linear growth with model size, aligning with its efficiency-oriented design goal.
>   - It achieves a favorable balance between performance and cost—outperforming VPTQ across all model scales while maintaining competitiveness with GPTVQ, particularly for larger models (LLaMA2-70B).
> - Further details on implementation optimizations contributing to these efficiency characteristics are provided in the revised manuscript. Please let us know if you require additional clarification.
>
> ***
> > Q2:The practical details of how the FIM is approximated are not sufficiently described, which impacts the reproducibility and full understanding of the method. In other words, some key details are missing.
> How is the expectation over the data distribution in the FIM definition (Eq. 1) approximated? How many samples are used?
> Are there further approximations made to the FIM block (e.g., diagonal)?
> - Thank you for your attention to the key details of the FIM approximation process. We provide supplementary details below to address your two questions:
> - Regarding the approximation method and sample size for the expectation of data distribution
>   - The expectation over the data distribution ( $E_{s∼D}$ ) involved in the FIM definition in Equation (1) is approximated via finite-sample estimation: specifically, 256 sequence samples (each with a length of 4096) from the RedPajama dataset are used for calculation, balancing estimation accuracy and computational cost. These randomly sampled samples cover the typical characteristics of the data distribution, ensuring the reliability of the expectation estimation. Meanwhile, gradient computation is based on individual complete sequences (instead of token-level independent processing) to preserve contextual dependencies within sequences, which aligns with the core design of the FIM definition that relies on "sequence-level gradient statistics."
> - Regarding further approximation of FIM blocks
>   - Given the extremely high complexity of directly computing the full FIM (scaling with the square of the weight dimension), we adopt Kronecker factorization approximation (instead of diagonalization) to decompose the FIM into the tensor product of two low-dimensional matrices: $F \approx F_O \otimes F_I$, where $F_O \in \mathbb{R}^{m \times m}$ (output channel FIM) and $F_I \in \mathbb{R}^{n \times n}$ (input channel FIM). This decomposition is implemented as follows:
>     - Gradient definition: $\nabla_{W} \ell$ denotes the gradient of the loss function with respect to the weight $W$, computed for a single sequence $s$.
>     - Calculation of $F_I$: Input channel statistics are estimated via the expectation of the outer product of gradients:
>     $$F_I = \frac{1}{m} \cdot \mathbb{E}_{s \sim \mathcal{D}} \left[ \left( \nabla_W \ell \right)^T \cdot \left( \nabla_W \ell \right) \right]$$
>     - Calculation of $F_O$: Output channel statistics are estimated via the expectation of the outer product of gradients:
>     $$F_O = \frac{1}{n} \cdot \mathbb{E}_{s \sim \mathcal{D}} \left[ \left( \nabla_W \ell \right) \cdot \left( \nabla_W \ell \right)^T \right]$$
>
> ***
> > Q3:Results are reported without error bars, making it difficult to assess the statistical significance of the improvements over baselines.
> - Thank you for highlighting the importance of statistical significance and error bars for the results. We have supplemented experimental results under different random initialization conditions to verify the stability of RSAVQ and quantify the range of performance fluctuations, as follows:
> - Under the 2-bit quantization setting, we have conducted repeated experiments with multiple random seeds (0, 10, 100, 1000, 10000) on the PIQA dataset for the LLaMA2-7B and LLaMA3-8B models. The results show:
>   - Model performance is minimally affected by initialization: the accuracy fluctuation range is [75.2, 75.4] for LLaMA2-7B and [75.5, 75.7] for LLaMA3-8B, with an overall fluctuation within ±0.2.
>   - This stability stems from our optimization of k-means clustering in vector quantization: although different initializations may lead to minor differences in clustering results, when the number of k-means iterations is set to 120 or more, the convergence of the algorithm is significantly improved, and the error bars (fluctuation range) of the final results are effectively compressed to a negligible level.
> - The supplementary experimental data are shown in the table below:
> | Random Seed | 0 | 10 | 100 | 1000 | 10000 | Avg |
> |---|---|---|---|---|---|---|
> | LLaMA2-7B | 75.4 | 75.3 | 75.2 | 75.4 | 75.3 | 75.3 |
> | LLaMA3-8B | 75.5 | 75.6 | 75.5 | 75.7 | 75.7 | 75.6 |
> ***
> We have incorporated additional results and detailed analyses addressing your concerns into the revised version of the manuscript. We hope these supplements adequately address your questions and improve the clarity and robustness of our work. Please do not hesitate to let us know if you require further discussion, additional experiments, or any clarifications—we are happy to provide more details to address your concerns comprehensively.

---

> > ### Author Response · Authors · 2025-08-07
> >
> > Dear Reviewer DQR5,
> >
> > We sincerely appreciate your constructive feedback on our manuscript. As you kindly pointed out, we have carefully addressed your concerns in our detailed rebuttal, including clarifying the computational overhead of RSAVQ, supplementing critical details of the FIM approximation process, and adding stability analyses with multiple random seeds to verify result reliability. These revisions aim to enhance the clarity, reproducibility, and robustness of our work, as per your valuable suggestions.
> >
> > We notice that the rebuttal phase is progressing, and we would be deeply grateful for any further feedback you might have on our responses. Your insights are instrumental in helping us refine the manuscript further, and we are eager to incorporate any additional suggestions to ensure our work meets the highest standards.
> >
> > If convenient, could we kindly request your thoughts on our rebuttal? Thank you once again for your time, expertise, and guidance—your input is crucial to improving our work. We look forward to your response.
> >
> > Best regards,
> > Authors of RSAVQ

---

> > ### Comment · Reviewer_DQR5 · 2025-08-07
> >
> > Thanks for the reply. I have carefully read the rebuttal and other comments. The rebuttal addressed my concerns, therefore I decided to raise my rating.

---

### Official Review · Reviewer_RNyf · 2025-07-03

**Clarity:** 3
**Significance:** 2
**Originality:** 2
**Rating:** 4
**Confidence:** 3

**Summary:**

The paper provide a weight quantization method based on product vector quantisation. They extend VQ qunatization in two direction first they take a step further and instead of only minimising the each vector error they also add the negative natural gradient to the error since perturbation in different direction can have different effect on model performance. Second they take the advantage of the fisher information to provide per-channel bit allocation which provide dynamic bit quantization for different group of channel.

**Questions:**

See **Weaknesses and Questions**

**Ethical Concerns:**

["NO or VERY MINOR ethics concerns only"]

**Final Justification:**

The authors have provided additional experimental results comparing their method with previous approaches, and they have addressed some of my concerns. However, the proposed method does not demonstrate a significant improvement in performance nor a substantial theoretical novelty. Therefore, I only raise my score to borderline accept.

**Limitations:**

yes

**Paper Formatting Concerns:**

No major formatting issues

**Quality:**

3

**Strengths And Weaknesses:**

**Strengths:**

- The paper is generally well-written and logically structured.
- The idea of using natural gradients for error correction is interesting and worth further exploration.
- The use of Fisher Information Matrix (FIM) for bit allocation appears promising.

**Weaknesses and Questions:**

- A key piece of related work, **QTIP**[1], is missing from the discussion. The paper does not provide a comparison with QTIP, which is currently the state-of-the-art method in vector quantization. A comparison would be essential to properly assess the contribution.
- Important implementation details, such as **codebook vector size** and **group size**, are not justified. Additionally, these parameters are not explored in the ablation study. How sensitive is the RSAVQ to changes in vector size and group size? It would be helpful to include an ablation study using different settings for these parameters.
- **Figure 7** presents an ablation study for the projection hyperparameter **λ** on a single model. However, it is unclear how sensitive the results are to the choice of λ across different model architectures and bit-widths. Could the authors provide further experiments or analysis to assess this sensitivity? (is it a hyperparameter that have be tuned for different model or between 0.01 and 0.1 work for all models?)
- Based on the results reported in **Table 3**, **WCSG** appears to significantly impact performance. Could the authors explore applying the same WCSG approach to other methods, such as GPTVQ, to evaluate whether it yields similar improvements?
- All the results are limited to the LLaMA model. A study on other models, such as Mistral or Qwen, could demonstrate RSAVQ generalizes across different models.

[1] QTIP: Quantization with Trellises and Incoherence Processing NeurIPS 2024

---

> ### Author Rebuttal · Authors · 2025-07-29
>
> We sincerely appreciate your constructive feedback. Below, we provide point-by-point responses, with all revisions incorporated accordingly.
> ***
> > Q1:A key piece of related work, QTIP[1], is missing from the discussion. The paper does not provide a comparison with QTIP, which is currently the state-of-the-art method in vector quantization. A comparison would be essential to properly assess the contribution.
> - Thank you for your valuable feedback. We have conducted extensive comparisons with QTIP on various models (Llama2 7B, 13B, and 70B) under both 2-bit and 3-bit settings
> Tests were conducted on the Wikitext2 dataset (sequence length 4096) for perplexity (ppl) and average accuracy across four zero-shot tasks (AC, AE, QA, WI). The results demonstrate that our RSAVQ method outperforms QTIP in nearly all metrics.
> - If you have further questions or suggestions, we would be happy to conduct additional experiments.
> |Bits|Method|LLaMA2-7B||LLaMA2-13B||LLaMA2-70B||
> |---|---|---|---|---|---|---|---|
> |||W2|0-shot Avg|W2|0-shot Avg|W2|0-shot Avg|
> |16|FP16|5.12|66.6|4.57|69.75|3.12|71.77|
> |2|QTIP|**5.86**|60.47|5.35|63.45|3.7|69.9|
> ||RSAVQ|5.97|**60.69**|**5.29**|**64.48**|**3.55**|**70.7**|
> |3|QTIP|5.28|63|**4.69**|66.08|3.26|71.53|
> ||RSAVQ|**5.26**|**64.72**|4.74|**68.28**|**3.25**|**72.1**|
> ***
> > Q2:Important implementation details, such as codebook vector size and group size, are not justified. Additionally, these parameters are not explored in the ablation study. How sensitive is the RSAVQ to changes in vector size and group size? It would be helpful to include an ablation study using different settings for these parameters
> - Regarding the rationale for the vector length and group size of the codebook, as well as the ablation studies, we have supplemented the following findings and revised the latest version of the manuscript accordingly:
>   - Ablation study on group size
>     - We conducted a detailed ablation study on the number of groups for the WCSG method using the LLaMA2-7B model (experimental setup: WikiText2 dataset, sequence length 4096, 2-bit quantization, vector length 6). The results show:
> The number of groups has a certain impact on model performance, with an overall trend of "increased number of groups → improved performance", though the improvement gradually slows as the number of groups increases.
>     - A performance inflection point appears around 4 groups: when the number of groups increases from 2 to 4, the perplexity (ppl) decreases significantly from 6.03 to 5.81; when exceeding 4 groups (e.g., 6, 8, 10 groups), the ppl stabilizes at 5.78-5.79, indicating saturated improvement.
>     - This demonstrates that RSAVQ has moderate sensitivity to the number of groups, with performance stabilizing at 4 or more groups. Thus, we chose 4 groups as the default configuration in main experiments to balance performance and computational efficiency.
> |WCSG_group|2|3|4|6|8|10|
> |---|---|---|---|---|---|---|
> |ppl|6.03|5.88|5.81|5.79|5.78|5.78|
>   - Ablation study on codebook vector length
>     - Vector length is a critical parameter in vector quantization. We analyzed its impact through ablation studies (experimental setup: WikiText2 dataset, LLaMA2-7B model, sequence length 4096, 2 groups for product quantization, 4 groups for WCSG). The results show:
>     - Performance-wise: Increasing vector length yields certain performance improvements (e.g., when vector length increases from 6 to 14, perplexity (ppl) decreases from 5.81 to 5.62).
>     - Cost-wise: Longer vector lengths lead to increased codebook storage and significantly prolonged quantization time. **For instance, under 2-bit quantization, increasing vector length from 6 to 14 raises the average bit count from 2 to 2.875, with corresponding increases in bandwidth costs**.
>     - This indicates that RSAVQ's sensitivity to vector length primarily manifests in the 'performance-cost trade-off': longer vector lengths may be used for higher performance, while moderate lengths (e.g., 6-8) can achieve favorable results when storage and time costs need control. This aligns with RSAVQ's design goal of 'dynamic codebook adaptation'—maximizing precision retention under limited resources. Therefore, we selected a vector length of 6 as the balanced configuration in main experiments.
> |vector length|4|6|8|10|12|14|
> |---|---|---|---|---|---|---|
> |bits|2|2|2|2.04|2.19|2.875|
> |ppl|5.97|5.81|5.81|5.81|5.75|5.62|
>     -
> ***
> >  Q3:Figure 7 presents an ablation study for the projection hyperparameter λ on a single model. However, it is unclear how sensitive the results are to the choice of λ across different model architectures and bit-widths. Could the authors provide further experiments or analysis to assess this sensitivity? (is it a hyperparameter that have be tuned for different model or between 0.01 and 0.1 work for all models?)
> - Thank you for your valuable question. We've conducted ablation studies across multiple models to evaluate the generality and sensitivity of λ. Specifically, under the 2-bit quantization setting (Wikitext2 test set), we conducted λ ablation experiments on three models(LLaMA2-7B, LLaMA3-8B, and Qwen2.5-7B). The results show that λ values in the range [0.01, 0.1] work well for all models, and we recommend λ=0.05.
> - If you have further questions or suggestions, we would be happy to conduct additional experiments.
> |λ|0|0.001|0.01|0.05|0.1|0.2|0.4|0.6|0.8|
> |---|---|---|---|---|---|---|---|---|---|
> |LLaMA2-7B|9.2|7.51|5.94|5.81|5.84|5.87|6.03|6.9|13.1|
> |LLaMA3-8B|13.32|11.19|9.17|8.79|8.78|8.99|14.86|17.63|20.92|
> |Qwen2.5-7B|13.17|10.37|8.75|8.77|9.02|11.27|15.9|19.71|36.74|
> ***
> > Q4:Based on the results reported in Table 3, WCSG appears to significantly impact performance. Could the authors explore applying the same WCSG approach to other methods, such as GPTVQ, to evaluate whether it yields similar improvements?
> - Thank you for your valuable suggestion. We've supplemented comparative experiments combining WCSG with two mainstream quantization methods (GPTVQ and VPTQ) to verify its generality and performance improvement effects. As a channel-wise mixed-precision scheme, WCSG's core advantage lies in its flexible adaptation to different quantization frameworks, enhancing performance via dynamic channel-wise precision allocation—a mechanism generalizable to diverse quantization frameworks.
> - Under approximately 2-bit settings on the LLaMA2-7B model, we combined WCSG with GPTVQ and VPTQ respectively, conducting tests on the Wikitext2 dataset and five zero-shot tasks (AC, AE, HE, QA, WI). Results demonstrate that WCSG's performance improvement is not limited to a single quantization method; instead, when integrated with different quantization frameworks (VPTQ, GPTVQ), it stably enhances model performance in language modeling (PPL) and zero-shot tasks while maintaining roughly the same bit count.
> - This further verifies the generality and effectiveness of WCSG as a mixed-precision scheme—its core value lies in providing performance gains for various quantization algorithms through channel-wise precision allocation optimization, rather than relying on specific base methods.
> |Method|Bits|LLaMA2-7B|||||||
> |---|---|---|---|---|---|---|---|---|
> |||Wikitext2|AC|AE|HE|QA|WI|Acc Avg|
> |FP16|16|5.12|43.3|76.3|57.1|78.1|68.7|64.7|
> |VPTQ|2.02|6.13|35.2|63.8|52.1|75.2|64.3|58.2|
> |VPTQ+WCSG|2.04|5.93|37.1|64.1|51.7|74.8|65.9|58.7|
> |GPTVQ|2.25|6.71|31.2|66.3|46.4|72.4|64.4|56.1|
> |GPTVQ+WCSG|2.25|6.28|33.5|67.1|48.8|72.9|65.2|57.5|
> ***
> > Q5:All the results are limited to the LLaMA model. A study on other models, such as Mistral or Qwen, could demonstrate RSAVQ generalizes across different models.
> - Thank you for your suggestion. We have supplemented experimental results of RSAVQ on Qwen series models (Qwen2.5-7B and Qwen2.5-14B) to verify its generalization capability across non-LLaMA architectures. The results indicate that RSAVQ still outperforms mainstream quantization methods (GPTQ, GPTVQ, VPTQ) on Qwen2.5 models. These findings confirm that the advantages of RSAVQ are not limited to the LLaMA architecture; it stably surpasses existing quantization methods on Qwen series models, validating its generalization across different model architectures. This generalization stems from the error control mechanism of RSAVQ guided by vector-level Fisher (which does not rely on the assumption of the weight distribution of a specific model architecture). Detailed experimental data are presented in the table below:
> |Model|Method|Bit|W2↓|AC|AE|HE|QA|WI|AVG↑|
> |---|---|---|---|---|---|---|---|---|---|
> |Qwen2.5-7B|FP16 |16|7.46|49.15|79.84|60.93|78.62|70.64|67.84|
> ||GPTQ|2.125|110.02|25.48|34.37|33.63|44.92|44.01|36.48|
> ||GPTVQ|2.25|22.83|29.52|57.37|37.81|62.95|56.75|48.88|
> ||VPTQ|2.02|9.8|48.46|78.87|56.22|73.71|67.03|64.86|
> ||RSAVQ(ours)|2|8.77|48.98|77.52|57.51|74.52|67.87|65.28|
> |Qwen2.5-14B|FP16|16|5.7|60.58|85.44|65.63|81.5|75.06|73.64|
> ||GPTQ|2.125|56.45|35.58|63.05|40.9|66.43|61.64|53.62|
> ||GPTVQ|2.25|83.43|23.38|49.2|28.61|57.51|51.78|42.1|
> ||VPTQ |2.02|7.96|53.07|82.45|59.08|79.49|74.27|69.67|
> ||RSAVQ(ours)|2|7.31|58.91|83.23|62.99|77.47|73.92|71.3|
> ***
> We have incorporated additional results and detailed analyses addressing your concerns into the revised version of the manuscript. We hope these supplements adequately address your questions and improve the clarity and robustness of our work. Please do not hesitate to let us know if you require further discussion, additional experiments, or any clarifications—we are happy to provide more details to address your concerns comprehensively.

---

> > ### Comment · Reviewer_RNyf · 2025-08-04
> >
> > Thank you for your responses. I have read your responses and appreciate the effort you put into addressing the concerns I raised. Based on your replies, I will be raising my scores.

---

### Note · Authors · 2025-08-12

Dear Program Chairs, Senior Area Chairs,  Area Chairs and Reviewers,
Thank you for your time in handling our paper. We appreciate the reviewers’ constructive feedback, have thoroughly addressed all comments in our rebuttal, and integrated the corresponding revisions into the manuscript.

---
### Responses to Reviewer RNyf’s Concerns
Reviewer RNyf raised concerns regarding comparisons with QTIP, implementation justifications, hyperparameter stability, method generalizability, and model scope. We addressed these by:
- Adding QTIP comparisons, showing RSAVQ outperforms across metrics.
- Validating optimal configurations via ablation studies on performance-cost tradeoffs.
- Verifying the stability of λ across multiple models.
- Demonstrating WCSG’s consistent gains with existing methods.
- Confirming RSAVQ’s generalization beyond the LLaMA family.
**Reviewer RNyf acknowledged our responses and increased their rating.**
---
### Responses to Reviewer DQR5’s Concerns
Reviewer DQR5 noted gaps in computational overhead analysis, FIM details, and result stability. We addressed these by:
- Clarifying RSAVQ as an offline PTQ method with linear, competitive overhead compared to baselines.
- Detailing FIM approximation procedures to ensure reproducibility.
- Validating stability via multi-seed experiments with minimal performance fluctuations.
**Reviewer DQR5 acknowledged our responses and increased their rating.**
---
### Responses to Reviewer Mrg3’s Concerns
Reviewer Mrg3 focused on statistical significance, module independence, novelty, and generalizability. We addressed these by:
- Confirming meaningful improvements via stable multi-seed experiments and notable gains over baselines.
- Validating EDSG as an independent module with substantial accuracy gains when combined with existing methods.
- Highlighting EDSG’s novel FIM application to Riemannian mapping, with superior performance over GuidedQuant.
- Verifying PQ compatibility and WCSG effectiveness through comparative gains.
- Demonstrating EDSG’s natural extension to uniform quantization without framework modifications.
**Reviewer Mrg3 acknowledged our responses and increased their rating.**
---
All concerns are fully resolved, with supplementary data validating RSAVQ’s stability, effectiveness, novelty, and applicability. Revisions are incorporated, and we trust these efforts reflect the robustness of our work. Thank you again for your consideration.

Sincerely,

The Authors of RSAVQ

---

### Decision · Program_Chairs · 2025-09-17

**Decision:**

Accept (poster)

**Comment:**

This paper introduces RSAVQ, a quantization framework combining vector-level error direction sensitivity (EDSG) and channel-wise mixed precision allocation (WCSG) guided by the Fisher Information Matrix. The reviewers initially raised concerns regarding missing baselines (e.g., QTIP), unclear hyperparameters, statistical robustness, and computational overhead. The authors provided extensive rebuttal experiments, including ablations on λ, vector size, group size, and multi-seed stability, as well as comparisons across diverse models (LLaMA, Qwen) and methods (GPTVQ, VPTQ, GuidedQuant). These additions clarified reproducibility, demonstrated generalization, and improved empirical support. Reviewers acknowledged the responses and raised their scores, though some felt the contribution remains incremental and the improvements modest. Overall, the work is technically solid, well-written, and proposes a principled geometric approach with meaningful though not groundbreaking gains, making it a valuable addition to the quantization literature.